# Censored Quantile Regression Neural Networks for Distribution-Free Survival Analysis

**Tim Pearce**[1,2,*]**, Jong-Hyeon Jeong**[3]**, Yichen Jia**[3]**, Jun Zhu**[1*]
[1]Dept. of Comp. Sci. & Tech., NRist Center, Tsinghua-Bosch Joint ML Center, Tsinghua University
[2]Microsoft Research, [3]University of Pittsburgh

## Abstract

This paper considers doing quantile regression on censored data using neural networks (NNs). This adds to the survival analysis toolkit by allowing direct prediction of the target variable, along with a distribution-free characterisation of uncertainty, using a flexible function approximator. We begin by showing how an algorithm popular in linear models can be applied to NNs. However, the resulting procedure is inefficient, requiring sequential optimisation of an individual NN at each desired quantile. Our major contribution is a novel algorithm that simultaneously optimises a grid of quantiles output by a single NN. To offer theoretical insight into our algorithm, we show firstly that it can be interpreted as a form of expectation-maximisation, and secondly that it exhibits a desirable 'self-correcting' property. Experimentally, the algorithm produces quantiles that are better calibrated than existing methods on 10 out of 12 real datasets.
Code: https://github.com/TeaPearce/Censored_Quantile_Regression_NN.

## 1 Introduction

Domains such as biomedical sciences and reliability engineering often produce datasets with a particular challenge – for many datapoints the target variable is not directly observed and only a lower or upper bound is recorded. For example, when modelling the time to failure of a machine, in many cases it might have been preemptively maintained before a failure event was observed, meaning only a lower bound on the time to failure is recorded. This is known as censored data, which is studied in the field of survival analysis.

Recently, there has been interest in combining ideas from survival analysis with neural networks (NNs), in the hope of leveraging the capabilities of deep learning for this important class of problem. Quantile regression has proven valuable in survival analysis, since it directly predicts the variable of interest, naturally capturing the uncertainty of the conditional distribution with no assumptions made about the distribution of residual errors [Peng, 2021]. Despite a wealth of research in the linear setting, explorations into its combination with NNs are at an early stage [Jia and Jeong, 2022]. This paper advances this line of research, showing how an estimator from Portnoy [2003], used for doing quantile regression in linear models on left and right censored data, can be combined with NNs.

Section 2 introduces Portnoy's estimator, and outlines a sequential grid algorithm that is used in its optimisation for linear models [Neocleous et al., 2006]. As our first contribution, Section 3.1 shows how this algorithm can be directly adapted to work with NNs. Unfortunately, the resulting method is inefficient, requiring sequential optimisation of a new NN at each predicted quantile. Section 3.2 outlines our main contribution; an improved algorithm that is well suited to NNs, named 'censored quantile regression neural network' (CQRNN). Our CQRNN algorithm is a significant departure from

---

*Majority of work completed while Tim Pearce was at Tsinghua University. Correspondence to: tim.pearce@microsoft.com; dcszj@tsinghua.edu.cn.

the sequential approach, and we offer theoretical insights into it in Section 4, interpreting it as a form of expectation-maximisation (EM), and also analysing a 'self-correcting' property. In Section 6 the algorithm's effectiveness is empirically demonstrated by benchmarking against alternative methods on simulated and real data. Careful ablations illuminate effects of the algorithm's hyperparameters.

## 2 Background

We first introduce censored data. Consider a dataset $\mathcal{D} = \{\{\mathbf{x}_1, y_1, \Delta_1\}, ..., \{\mathbf{x}_N, y_N, \Delta_N\}\}$, where $\mathbf{x}_i \in \mathbb{R}^D$ is the input, $y_i \in \mathbb{R}$ is the (possibly censored) variable to regress on, and $\Delta_i$ represents the observed/censored indicator. We assume two data generating distributions, one for the target variable, $t_i \sim p_t(t|\mathbf{x}_i)$, and another for the censoring variable, $c_i \sim p_c(c|\mathbf{x}_i)$. In the case of right censoring, we only observe the smaller of these two, which forms our dataset labels, $y_i = \min(t_i, c_i)$, and $\Delta_i = \begin{cases} 0 & \text{if censored, } c_i < t_i \\ 1 & \text{else} \end{cases}$ . Whilst this paper concentrates on the right censored case, for all estimators and algorithms discussed, left censoring can be handled by simply inverting the labels, i.e. using $-y_i \ \forall i$ [Koenker, 2022].

In this paper we make common assumptions about $p_t(t|\mathbf{x}_i)$ and $p_c(c|\mathbf{x}_i)$; the target is independent of censoring given the covariates, $t \perp c \mid \mathbf{x}$ (sometimes termed 'random' censoring), but both $t$ and $c$ depend on $\mathbf{x}$ in different and possibly non-linear ways. These are more general than some alternative assumptions, such as using fixed-value censoring, $c_i = \text{constant } \forall i$ (e.g. [Powell, 1986]) or requiring the censoring distribution to be independent of $\mathbf{x}$ (e.g. [Jia and Jeong, 2022]).

### 2.1 Quantile Regression without Censoring

Ignoring censoring for a moment, the conditional quantile function is given by,

$$Q(\tau|\mathbf{x}_i) = \inf\{y_{i,\tau} : p(y_i \le y_{i,\tau}|\mathbf{x}_i) = \tau\}, \tag{1}$$

where, $\tau \in (0, 1)$. To learn such a function for a single target quantile, $\tau$, quantile regression minimises a 'checkmark' loss, $\rho_\tau(\cdot, \cdot) : \mathbb{R} \times \mathbb{R} \to \mathbb{R}$, [Koenker and Bassett, 1978],

$$\mathcal{L}_{\text{check}}(\theta, \mathcal{D}, \tau) := \frac{1}{N} \sum_{i=1}^{N} \rho_\tau(y_i, \hat{y}_{i,\tau}), \tag{2}$$

$$\rho_\tau(y_i, \hat{y}_{i,\tau}) := (y_i - \hat{y}_{i,\tau})(\tau - \mathbb{I}[\hat{y}_{i,\tau} > y_i]), \tag{3}$$

for model parameters, $\theta$, and a prediction, $\hat{y}_{i,\tau} = \psi_\tau(\mathbf{x}_i, \theta)$, made by some model $\psi_\tau : \mathbb{R}^D \to \mathbb{R}$, with $\mathbb{I}[\cdot]$ as the indicator function. A linear model refers to when, $\hat{y}_{i,\tau} = \theta_\tau^\intercal \mathbf{x}_i$, with, $\theta_\tau \in \mathbb{R}^D$.

### 2.2 Portnoy's Censored Quantile Regression Estimator

Directly optimising the loss in Eq. 2 can produce undesirable models when a dataset contains censored observations. Naively ignoring censoring indicators or excluding all censored data will, in the general case, induce unfavourable bias [Zhong et al., 2021]. The algorithms proposed in this paper use a re-weighting scheme introduced in Portnoy [2003] that does account for censoring. Define two disjoint sets of indices, one for censored and one for observed datapoints, $\mathcal{S}_{\text{censored}}$ and $\mathcal{S}_{\text{observed}}$, letting, $N_c := |\mathcal{S}_{\text{censored}}|$ and $N_o := |\mathcal{S}_{\text{observed}}|$. Portnoy's estimator minimises,

$$\mathcal{L}_{\text{Port.}}(\theta, \mathcal{D}, \tau, \mathbf{w}, y^*) = \sum_{i \in \mathcal{S}_{\text{observed}}} \rho_\tau(y_i, \hat{y}_{i,\tau}) + \sum_{j \in \mathcal{S}_{\text{censored}}} w_j \rho_\tau(y_j, \hat{y}_{j,\tau}) + (1 - w_j)\rho_\tau(y^*, \hat{y}_{j,\tau}). \tag{4}$$

While the loss for observed datapoints is unchanged, censored datapoints have been split into two 'pseudo' datapoints – one at the censored value, and one at some large value, $y^* \gg \max_i y_i$ (Section 6.4 discusses $y^*$ further). Weight is apportioned between each pair of pseudo datapoints by,

$$w_j = \frac{\tau - q_j}{1 - q_j}, \tag{5}$$

where $q_j$ is the quantile at which the datapoint was censored, with respect to the target value distribution, i.e. $p_t(t_j > c_j|\mathbf{x}_j)$. We use $\mathbf{w} \in \mathbb{R}^{N_c}$ and $\mathbf{q} \in \mathbb{R}^{N_c}$ to denote the vector of all weights and quantiles respectively, indexed as $w_j$ and $q_j$. Given these weights, Portnoy's estimator has been shown to be analogous to the Kaplan-Meier (KM) estimator [Portnoy, 2003].

**Algorithm 1** Sequential grid algorithm for NNs.

---

**Require:** Dataset $\mathcal{D}$, $M$ parametric models $\psi_\tau(\cdot)$ each with randomly initialised parameters $\theta_\tau$, ordered quantiles to be estimated $\mathrm{grid}_\tau$, learning rate $\alpha$, pseudo y value $y^*$.

---

$\mathcal{S}_{\text{censored}} \leftarrow \{i \in \{0, 1, \ldots, N\} : \Delta_i = 0\}, \mathcal{S}_{\text{observed}} \leftarrow \{i \in \{0, 1, \ldots, N\} : \Delta_i = 1\}, K_{\text{cross}} \leftarrow \emptyset$
**for** $i = 0$ **to** $M - 1$ **do**
    $\tau \leftarrow \mathrm{grid}_\tau[i]$
    **if** $i = 0$ **then**                                         ▷ Initialise quantile estimates to zero
        $\hat{\mathbf{q}} \leftarrow \mathrm{zeros}(N_c)$
    **else**              ▷ Find indices which have been crossed and update their estimated quantiles
        $K \leftarrow \{j \in \mathcal{S}_{\text{censored}} : \psi_{\mathrm{grid}_\tau[i-1]}(\mathbf{x}_j) \le y_j \cap \psi_{\mathrm{grid}_\tau[i]}(\mathbf{x}_j) > y_j\}$
        $\hat{\mathbf{q}}[K] \leftarrow \mathrm{grid}_\tau[i-1]$
        $K_{\text{cross}} \leftarrow K_{\text{cross}} \cup K$
        $\hat{\mathbf{q}}[\neg K_{\text{cross}}] \leftarrow \tau$                          ▷ Sets $\hat{\mathbf{w}} = 0$ for uncrossed datapoints
    $\hat{\mathbf{w}} \leftarrow (\tau - \hat{\mathbf{q}})/(1 - \hat{\mathbf{q}})$
    $\theta \leftarrow \arg\min_{\theta_\tau} \mathcal{L}_{\text{Port.}}(\theta_\tau, \mathcal{D}, \tau, \hat{\mathbf{w}}, y^*)$              ▷ Fully optimise Eq. 4

---

## 2.3 Sequential Grid Algorithm

A challenge with Portnoy's estimator is that it creates a circularity. If the true quantiles $\mathbf{q}$ and hence $\mathbf{w}$ for all censored datapoints were known, Eq. 4 could be optimised straightforwardly, but the very reason to perform this optimisation is to estimate such quantiles! Prior work has tackled this issue in various ways (see Section 5). Here we discuss the most widely-used algorithm. It was originally presented in Portnoy [2003] and slightly modified in Neocleous et al. [2006], we refer to it as the 'sequential grid algorithm'.

Described in Algorithm 1, it sequentially steps through a grid of $M$ desired quantiles, arranged in strictly increasing order, and typically evenly spaced, e.g., $\mathrm{grid}_\tau = \{0.1, 0.2, \ldots, 0.8, 0.9\}$ for $M = 9$. A new model is fitted at each quantile. The algorithm terminates either when all quantiles in the grid have been iterated through, or if only censored datapoints lie above the current quantile. (Portnoy [2003] suggest handling the first quantile specially, but we have simplified this – see later.) Note we introduce $\hat{\mathbf{q}}$ & $\hat{\mathbf{w}}$ to explicitly designate model estimates of $\mathbf{q}$ & $\mathbf{w}$.

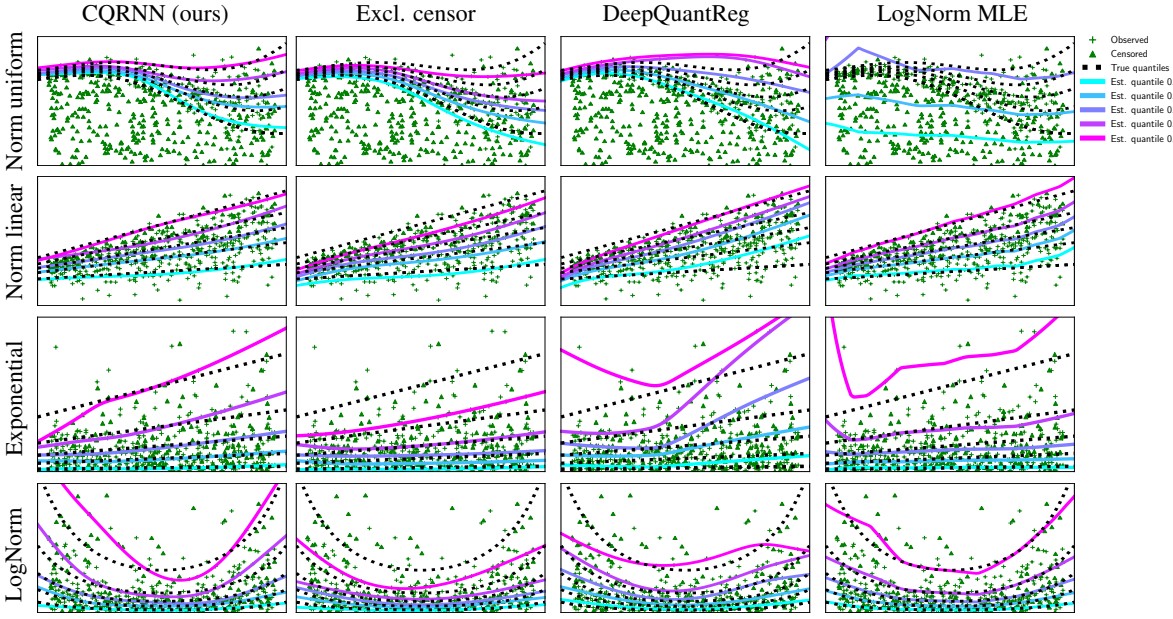

Figure 1: 1D synthetic datasets of varying functions (rows), fitted by various methods (columns). Estimated quantiles (blue through pink) compared to ground truth quantiles (dashed black lines). CQRNN recovers quantiles closest to the ground truth on most datasets.

# 3 Censored Quantile Regression and Neural Networks

This section considers applying the sequential grid algorithm to NNs, explaining why this is inefficient. It then describes the CQRNN algorithm, our main contribution, which avoids these pitfalls.

## 3.1 The Sequential Grid Algorithm and Neural Networks

To the best of our knowledge the sequential grid algorithm has been proposed and implemented only on linear models. Our first contribution is showing that, with three adjustments, it can be used in NNs.

Adj. 1) The $M$ models producing estimates, $\hat{y}_{j,\tau}$, are chosen be NNs, of any architecture with a single output. Adj. 2) While the linear version of the algorithm minimises Eq. 4 analytically with a parametric programming approach, for NNs this is instead minimised with stochastic gradient descent. Adj. 3) Portnoy [2003] proposed using a lengthy process for the first quantile. They set $\hat{\mathbf{w}} = \mathbf{1}$ and optimise Eq. 4. After optimisation, if any censored datapoints lie below the first quantile, these are removed from the dataset, and the procedure repeats until there are none. This repeated optimisation could go on for many iterations for large datasets and is costly for NNs. Instead, we simply set $\hat{\mathbf{q}} = \mathbf{0}$, and optimise the first quantile once only. We show this is roughly equivalent in Appendix A.2.

Unfortunately, this adaption is still inefficient. Inspecting Algorithm 1, one notes that $M$ NNs must be trained sequentially, meaning time complexity (both at training and at test) and memory complexity is $\mathcal{O}(M)$. This is not the case when doing quantile regression with NNs *without* censoring – a single NN typically uses multiple outputs, one per quantile, and their losses can be minimised simultaneously [Taylor, 2000]. For modest numbers of quantiles, this results in little overhead relative to a single NN, since the extra quantile outputs require only linear heads, while the bulk of the parameter count and computation time arise from the shared trunk of the NN.

Note that these inefficiencies are less problematic for linear models – optimisation of Eq. 4 can be performed quite rapidly, and memory complexity cannot obviously be improved beyond $\mathcal{O}(M)$. As such, it is the use of NNs that motivates the search for an alternative algorithm.

## 3.2 CQRNN Algorithm

Our proposed algorithm uses a single NN outputting a grid of quantile estimates that can be optimised simultaneously. This results in a more efficient algorithm in terms of training time, test time, and memory. Concretely, if the NN is a single hidden-layer NN with $H$ neurons, then the model is defined, $\psi_\tau(\mathbf{x}_j, \theta) = \theta_{1,\tau}^\mathsf{T} \phi(\theta_0^\mathsf{T} \mathbf{x}_i)$, with $\theta_0 \in \mathbb{R}^{D \times H}$ shared for all estimates and $\theta_{1,\tau} \in \mathbb{R}^{H \times 1}$ for the specific output head, and non-linearity, $\phi$.

The key insight behind our algorithm is that estimates of the censored weights, $\hat{\mathbf{w}}$, can be bootstrapped from the model *while* it learns, without requiring sequential optimisation. Intuitively, as the model trains, the estimates of the censored quantiles, $\hat{\mathbf{q}}$, and hence the censored weights, $\hat{\mathbf{w}}$, improve, and these feedback to create a more accurate loss function, allowing further improvement of the model.

We summarise the method in Algorithm 2. The quantile of a censored datapoint is estimated as whichever NN output is closest under the current model parameters. Whilst the algorithm is written in two steps, getting up-to-date estimates of censored quantiles, $\hat{q}_j$, requires only a forward pass, which is done at the point of loss optimisation anyway, so actually comes 'for free'. In our implementations, the algorithm is extended to leverage mini-batches and modern optimisers (e.g. Adam).

# 4 Analysis of CQRNN

The previous section introduced the CQRNN algorithm, motivating it intuitively, while this section considers its correctness. Providing precise analytical guarantees is very challenging, both due to the involvement of non-convex NNs, and the bootstrapping nature of the algorithm. We therefore consider more holistic ways of understanding and providing confidence in the method. We firstly show that it can be interpreted as a flavour of EM ('generalised hard' EM) – connecting to this well-studied class of algorithms provides some justification of CQRNN's design. We secondly validate the way the model bootstraps its own censored quantile estimates by describing a 'self-correcting' property.

**Algorithm 2** CQRNN algorithm.

---

**Require:** Dataset $\mathcal{D}$, one parametric model $\psi(\cdot)$ with randomly initialised parameters $\theta$ that can output $M$ quantile predictions, quantiles to be estimated $\mathrm{grid}_\tau$, learning rate $\alpha$, pseudo y value $y^*$.

$\mathcal{S}_{\text{censored}} \leftarrow \{i \in \{0, 1, \ldots, N\} : \Delta_i = 0\}, \mathcal{S}_{\text{observed}} \leftarrow \{i \in \{0, 1, \ldots, N\} : \Delta_i = 1\}$
**while not** convereged **do**
    *1. Hard expectation step*
    **for** $j \in \mathcal{S}_{\text{censored}}$ **do**
        $\hat{q}_j \leftarrow \arg\min_\tau |\hat{y}_{j,\tau} - y_j|$         ▷ Estimate quantile of censored data under current model
        $\hat{w}_j \leftarrow (\tau - \hat{q}_j)/(1 - \hat{q}_j)$
    *2. Partial maximisation step*
    $\theta \leftarrow \theta - \alpha \partial \mathcal{L}(\theta_\tau, \mathcal{D}, \tau, \hat{\mathbf{w}}, y^*)/\partial\theta \ \ \forall \tau \in \mathrm{grid}_\tau$         ▷ Gradient step to minimise Eq. 4

---

### 4.1 Interpretation as Expectation-Maximisation

EM is a two-stage iterative optimisation technique for finding maximum likelihood solutions. It can be useful when one has both observed and latent variables, and where knowing the latent variables would allow straightforward optimisation of a model's parameters and vice versa.

In our problem, we interpret the weights of the censored datapoints, $\mathbf{w}$, as latent variables, observed variables as $\mathbf{x}$ & $y$, and the NN parameters $\theta$ are to be optimised. Under our notation, the vanilla EM procedure [Bishop, 2006] can be written as follows (using a randomly initialising $\theta^{\text{old}}$),

1. Expectation step. Evaluate, $p(\mathbf{w}|\mathbf{x}, y, \theta^{\text{old}})$, to set up the expectation,

$$\mathcal{Q}(\theta, \theta^{\text{old}}) = \int_{\mathbf{w}} p(\mathbf{w}|\mathbf{x}, y, \theta^{\text{old}}) \log p(y, \mathbf{w}|\mathbf{x}, \theta) d\mathbf{w} = \mathbb{E}_{\mathbf{w}|\mathbf{x}, y, \theta^{\text{old}}}[\log p(y, \mathbf{w}|\mathbf{x}, \theta)]. \quad (6)$$

2. Maximisation step. Perform optimisation of the expected likelihood.

$$\theta^{\text{new}} = \arg\max_\theta \mathcal{Q}(\theta, \theta^{\text{old}}) \quad (7)$$

3. Set $\theta^{\text{old}} \leftarrow \theta^{\text{new}}$ and repeat until convergence.

We now show how our CQRNN method in Algorithm 2 can be interpreted as a flavour of EM, specifically generalised hard EM. The key steps in this interpretation are: 1) The likelihood for datapoints at each quantile are chosen to follow an asymmetric Laplace distribution. 2) The expectation step is treated as a hard assignment under the current NN parameters. 3) The likelihood is only partially maximised at each iteration.

**Likelihood Form**

A likelihood function is required for both the E & M steps. Theorem 1 shows how Eq. 4 can be interpreted as a negative log likelihood.

**Theorem 1.** *Let the likelihood for each datapoint at each quantile be an asymmetric Laplace distribution with scale, $\lambda = \sqrt{\tau - \tau^2}$, and asymmetry, $k = \tau/\sqrt{\tau - \tau^2}$. The negative log likelihood is,*

$$-\log p(y|\mathbf{x}, \theta, \mathbf{w}, y^*) = \sum_{\tau \in \mathrm{grid}_\tau} \mathcal{L}_{\text{Port.}}(\theta, y, \mathbf{x}, \tau, \mathbf{w}, y^*) + \text{constant} . \quad (8)$$

*Proof sketch.* The asymmetric Laplace distribution has been used in Bayesian treatments of quantile regression without censoring [Yu and Moyeed, 2001]. We extend this to Portnoy's loss with a weighted likelihood form using the censored weights. Appendix A provides the full proof.

**Expectation as a Hard Assignment of Latent Variables**

In some models, such as a Gaussian mixture model, the expectation step can be computed through analytical evaluation of the latent posterior distribution [Bishop, 2006]. In our case, $p(\mathbf{w}|\mathbf{x}, y, \theta^{\text{old}})$,

is intractable, though Eq. 4 depends linearly on each element $w_j$, so we need only consider the expectation, $\mathbb{E}_{w_j|\mathbf{x},y,\theta^{\text{old}}}[w_j]$. The algorithm can then be interpreted as making an assignment under the current model parameters,

$$\hat{\mathbb{E}}_{w_j|\mathbf{x}_j,y_j,\theta^{\text{old}}}[w_j] = \sum_{w_j} w_j \hat{p}(w_j|\mathbf{x}_j,y_j,\theta^{\text{old}}) = \frac{\tau - \hat{q}_j}{1 - \hat{q}_j}, \tag{9}$$

$$\hat{p}(w_j|\mathbf{x}_j,y_j,\theta^{\text{old}}) = \begin{cases} 1 \text{ if } w_j = \frac{\tau - \hat{q}_j}{1-\hat{q}_j}, \text{ where } \hat{q}_j = \arg\min_\tau |\hat{y}_{j,\tau} - y_j| \\ 0 \text{ else} \end{cases} . \tag{10}$$

This is a 'hard' assignment, where latents are assigned to the most likely values under the current model (e.g. datapoints are attributed to the nearest clusters in the K-means algorithm [Bishop, 2006]), and gives rise to a class of algorithms termed hard EM [Samdani et al., 2012].

Note that the in Eq. 6 we can rewrite, $\log p(y, \mathbf{w}|\mathbf{x}, \theta) = \log p(\mathbf{w})p(y|\mathbf{x}, \theta, \mathbf{w})$. Since we choose only a single setting for $\mathbf{w}$ in the outer expectation, this dependence disappears, and we require maximisation of, $\mathbb{E}_{\mathbf{w}|\mathbf{x},y,\theta^{\text{old}}}[\log p(y|\mathbf{x}, \theta, \mathbf{w})]$.

**Partial Maximisation**

A second departure from the standard EM algorithm, is that Algorithm 2 performs only a partial maximisation of the likelihood, taking a single gradient step. This has been shown to provide similar guarantees, and is termed 'generalised' EM [Neal and Hinton, 1998].

One could consider an alternative version of the algorithm that is closer to the standard EM procedure, where the estimated quantiles of censored data points are fixed while the maximisation step of NN parameters is run to convergence over multiple training epochs, before the estimates are updated in the expectation step, and repeating. In our case this is less effective – the maximisation requires running a forward pass through the NN, and obtaining estimated quantiles under current NN parameters once this is done is trivial, so we can access up-to-date estimates essentially for free. Appendix Figure 5 empirically demonstrates that CQRNN's partial maximisation approach produces the fastest convergence.

## 4.2 Self-Correcting Property

The CQRNN bootstraps weight estimates from the current model as it trains. It's not immediately obvious why this bootstrapping approach should converge to something sensible – what if bad initial weight estimates lead to worse ones? As a second insight into the CQRNN algorithm, Theorem 2 shows that when a censored weight, $\hat{w}_j$, is estimated incorrectly, the algorithm acts in a way to adjust the estimated quantiles in a favourable way – we refer to this as 'self-correcting'.

**Theorem 2.** *If $\hat{q}_j$ is underestimated, one iteration of the algorithm acts to increase the quantile predictions, $\hat{y}_{j,\tau}$, by the same amount, or even higher, than if the weight had been correct. If $\hat{q}_j$ is overestimated, $\hat{y}_{j,\tau}$, is decreased by the same amount, or even lower, than with the correct weight.*

*Proof sketch.* Denote $q_j$ the true quantile that censored datapoint $j$ is censored in, and $w_j$ the corresponding true weight. We derive the expression for the gradient wrt the predicted quantiles, $\hat{y}_{j,\tau}$, finding that if $\hat{w}_j$ is underestimated it holds that, $\frac{\partial \mathcal{L}_{\text{Port.}}(\theta,\mathcal{D},\tau,\hat{\mathbf{w}},y^*)}{\partial \hat{y}_{j,\tau}} \leq \frac{\partial \mathcal{L}_{\text{Port.}}(\theta,\mathcal{D},\tau,\mathbf{w},y^*)}{\partial \hat{y}_{j,\tau}}$, and hence gradient descent applies an adjustment in the desired direction of equal or greater magnitude than if the true weight had been used. The reverse holds for overestimated $\hat{w}_j$. Appendix A provides a full proof.

## 5 Related Work

**Survival analysis and NNs.** Following the widespread success of deep NNs over the past decade, there has been a wave of research applying NNs to survival analysis – for instance by modifying the CoxPH model [Katzman et al., 2018], or framing the task as ordinal classification [Lee et al., 2018]. Closer to our work are methods that use NNs to output parameters of a distribution such as the Weibull [Martinsson, 2016], seek robust training objectives for these models [Avati et al., 2019], or help with their optimisation [Tang et al., 2022]. Our work stands out as offering a way to

directly estimate the target variable at pre-specified quantiles, without enforcing any distributional assumption. A limitation is that the distribution may only be predicted at these quantiles (unless additional assumptions are made to allow interpolation between these).

**Quantile regression and NNs.** Quantile regression has proven an attractive option to enable NNs to move beyond point predictions. This allows quantification of a NN's aleatoric uncertainty [Tagasovska and Lopez-Paz, 2019]. It is attractive due to its straightforward implementation, and avoidance of any distributional assumption. For example, it has found use in reinforcement learning to capture the *distribution* of rewards, rather than just the mean [Dabney et al., 2017]. Since NNs are flexible function approximators, particular attention has been paid to the crossing quantile problem [Bondell et al., 2010, Zhou et al., 2020, Brando et al., 2022]. CQRNN borrows ideas from this line of work, and further tackles the challenge of learning quantiles under censored data.

**Censored quantile linear regression.** There is much work on censored quantile regression methods for linear models. Powell [1986] developed an estimator under fixed-value censoring which can be implemented with an algorithm from Fitzenberger [1997]. Portnoy [2003] developed an estimator under random censoring based on the KM estimator, while Peng and Huang [2008] developed an alternative based on the Nelson–Aalen (NA) estimator. KM and NA are closely related, and Portnoy and Peng's methods have been reported to offer similar empirical performance [Koenker, 2008]. Koenker [2022] provides all above methods in the popular 'quantreg' R package. Other notable methods include Yang et al. [2018], based on the data augmentation algorithm, and Wang and Wang [2009], whose estimator is similar to Portnoy's but utilises local estimates of the KM, computed with a kernel method. See Peng [2021] for a review of the broader area. Our work allows modelling of flexible non-linear quantile functions, leveraging the powerful representation learning abilities of NNs. Although, this sacrifices the interpretability of coefficients of linear models.

**Censored quantile regression and NNs.** Only a small amount of work has been done in this area. The 'qrnn' R package [Cannon, 2019] offers the ability to train NNs under fixed-value left censoring, adopting an idea from the linear setting [Friederichs and Hense, 2007, Cannon, 2011]. Huttel et al. [2022] explore our objective but assume censoring times, $c_i$, are available for both censored and uncensored data points (i.e. the censoring distribution is known). We do not require this assumption. DeepQuantReg [Jia and Jeong, 2022] tackles the same objective as this paper. They showed that improvements in quantile estimation can be obtained relative to naive methods. Their work differs from ours significantly – they base their method around an estimator for the median from Huang et al. [2007], requiring an assumption that the censoring distribution is independent of covariates.

# 6 Experiments

This section empirically investigates several questions. Q1) *How does the proposed CQRNN method compare with existing methods?* This is done qualitatively on synthetic 1D functions in Section 6.1 and quantitatively on synthetic and real datasets in Section 6.2. Q2) *How does the sequential grid algorithm compare to the CQRNN algorithm, both in terms of predictive accuracy and efficiency?* Explored in Section 6.3 Q3) *How is the CQRNN algorithm affected by its hyperparameters?* We investigate the impact of grid fidelity and $y^*$ in Section 6.4.

All experiments use fully-connected NNs with two hidden layers of 100 neurons, except for SurvM-NIST, when three convolutional layers are used. Grid size $M$ is set to 5, 9 or 19 depending on dataset size. Appendix B contains further details on hyperparameter settings, metrics, and datasets. Appendix C presents further results.

**Datasets.** We use three types of dataset. Type 1) *Synthetic target data with synthetic censoring.* Type 2) *Real target data with synthetic censoring.* Type 3) *Real target data with real censoring.* Whilst type 3 captures the kind of datasets we care about most, evaluation of quantiles is challenging since in real-world survival data the target conditional quantiles are not obtainable even at test time [Li and Peng, 2017]. In contrast, type 1 offers access to the ground truth quantiles for clean evaluation, but properties of these datasets may be less realistic. We introduce type 2 as a middle ground, which takes a real-world dataset without censoring, and synthetically overlays a censoring distribution to create training data. At test time, we have the option of not applying the censoring, providing samples from the clean target distribution, allowing clearer evaluation. We summarise all datasets in Table 2. Full details are in Appendix B.4. Our experiments exceed the number and variety of datasets used in popular recent works [Goldstein et al., 2020, Zhong et al., 2021]

Table 1: Performance of our proposed CQRNN algorithm (Algorithm 2) compared to the sequential grid method for NNs (Algorithm 1), over 200 random seeds.

| Dataset | Number quantiles | Training time speed up | Test time speed up | Parameter saving | TQMSE difference Seq. grid − CQRNN 95% conf. interval | CQRNN is sig. better in TQMSE? | Seq. grid is sig. better in TQMSE? | No statistical significant difference |
|---|---|---|---|---|---|---|---|---|
| Norm linear | 9 | 14.7× | 11.2× | 8.3× | -2.189 ± 0.245 | ✓ | | |
| Norm non-lin | 9 | 12.4× | 9.6× | 8.3× | -0.003 ± 0.001 | ✓ | | |
| Exponential | 9 | 12.4× | 8.0× | 8.3× | -0.093 ± 0.176 | | | ✓ |
| Weibull | 9 | 12.5× | 8.7× | 8.3× | -0.014 ± 0.016 | | | ✓ |
| LogNorm | 9 | 12.9× | 8.8× | 8.3× | -0.039 ± 0.028 | ✓ | | |
| Norm uniform | 9 | 12.6× | 8.1× | 8.3× | 0.175 ± 0.033 | | ✓ | |
| Norm heavy | 19 | 31.0× | 18.5× | 16.2× | 0.108 ± 0.211 | | | ✓ |
| Norm med. | 19 | 34.1× | 17.9× | 16.2× | -0.046 ± 0.004 | ✓ | | |
| Norm light | 19 | 31.7× | 19.2× | 16.2× | -0.035 ± 0.003 | ✓ | | |
| Norm same | 19 | 34.3× | 21.2× | 16.2× | -0.390 ± 0.049 | ✓ | | |
| LogNorm heavy | 19 | 33.6× | 19.9× | 16.2× | 0.005 ± 0.001 | | ✓ | |
| LogNorm med. | 19 | 31.3× | 13.1× | 16.2× | 0.019 ± 0.002 | | ✓ | |
| LogNorm light | 19 | 31.3× | 20.2× | 16.2× | -0.033 ± 0.004 | ✓ | | |
| LogNorm same | 19 | 29.8× | 18.4× | 16.2× | -0.371 ± 0.045 | ✓ | | |
| Total: | | | | | | 8/14 | 3/14 | 3/14 |

**Metrics.** Our objective is to measure how closely a model's predicted quantiles match those of the ground truth target distribution. We favour different metrics for each dataset type.

For type 1 datasets, since targets are generated synthetically it is possible to analytically compute the ground truth target quantile for an input $\mathbf{x}_i$, which we denote $y_{i,\tau}$. We compute the mean squared error (MSE) between the predictions and the ground truths across three quantiles, $\tau \in [0.1, 0.5, 0.9]$, (we ensure these are always present in $\text{grid}_\tau$). This is our first-choice metric when available. True quantile MSE (TQMSE) $:= \frac{1}{N} \sum_{\tau \in [0.1, 0.5, 0.9]} \sum_{i=1}^{N} (\hat{y}_{i,\tau} - y_{i,\tau})^2$.

In type 2 datsets *samples* from the uncensored target distribution are available but not the synthetic generating function. Our preferred metric is the checkmark loss across the three quantiles. Uncensored quantile loss (UQL) $:= \frac{1}{N} \sum_{\tau \in [0.1, 0.5, 0.9]} \sum_{i=1}^{N} \rho_\tau(y_i, \hat{y}_{i,\tau})$.

For datasets of type 3, we use two metrics. The concordance index (C-index) is computed using the median ($\tau = 0.5$). But this may not reveal anything about systematic bias of different models, nor about other quantiles $\tau \neq 0.5$. We secondly use censored D-Calibration (CensDCal) [Haider et al., 2020] which measures whether the empirical proportion of datapoints falling between pairs of consecutive quantiles, matches the target proportion, $\tau_{j+1} - \tau_j$. MSE of the deviation is then computed. There is also an uncensored version (UnDCal), which we can compute for type 1 & 2 datasets. Appendix B.2 gives further details.

**Baselines.** We compare against three methods that can be used to predict the quantiles of a target distribution using a NN. **Excl. Censor** – a method that naively excludes censored datapoints from the training data, optimising the loss in Eq. 3. **DeepQuantReg** – the only existing method in the literature proposing explicit output of quantiles from a NN on censored data [Jia and Jeong, 2022]. **LogNorm MLE** – A NN outputting parameters of a lognormal distribution, that's trained via maximum likelihood estimation (MLE) (details in Appendix B.3), and is a standard baseline in related work (e.g. [Avati et al., 2019, Goldstein et al., 2020]) since this distribution often suits properties of real-world time-to-event survival data [Kleinbaum and Klein, 2012].

### 6.1 Qualitative 1D Analysis

Figure 1 visualises the quantiles predicted by CQRNN and baseline methods for 1D datasets (Table 2 describes dataset functions, Figure 3 visualises further 1D datasets). Each dataset contains 500 datapoints drawn from, $\mathbf{x} \sim \mathcal{U}(0, 2)$ and $\text{grid}_\tau \in \{0.1, 0.3, 0.5, 0.7, 0.9\}$. CQRNN usually learns quantiles that are closer to the ground truth than baseline methods, particularly at higher quantiles. Excl. censor systemically makes underpredictions, and this worsens at higher quantiles since larger values of $y$ are more likely to be censored and excluded. DeepQuantReg avoids this systemic underprediction, but appears to introduce bias of its own. LogNorm MLE provides variable results – on Norm uniform it fails with excessive variance (an issue also observed by Avati et al. [2019]), but on LogNorm, since the target distribution matches the distribution output by the NN, it performs well.

## 6.2 Benchmarking

Our main experiment benchmarks CQRNN against all baselines across a wide variety of datasets, covering various domains, sizes, dimensionalities and censoring proportions (see Table 2). Some hyperparameter tuning was performed for each method (Appendix B.1). Figure 2 plots the preferred metric for each dataset type, though patterns were consistent across metrics – Table 4 provides a full breakdown.

For type 1 (synthetic) datasets, the quantitative results follow our qualitative observations, with CQRNN producing lowest TQMSE on all but the datasets generated from a Log Normal distribution, when it sometimes placed second behind LogNorm MLE.

CQRNN also produces the lowest UQL on all datasets of type 2. LogNorm MLE performs poorly on these, presumably since they are not typical time-to-event datasets which are well captured by the Log Normal distribution.

For type 3 datasets, CQRNN and LogNorm MLE usually perform best in terms of C-index, with error bars tending to overlap. In CensDCal, CQRNN consistently perform best, matched only by LogNorm MLE on two datasets, when error bars overlap. Excl. censor is the weakest method, with DeepQuantReg midway between it and CQRNN.

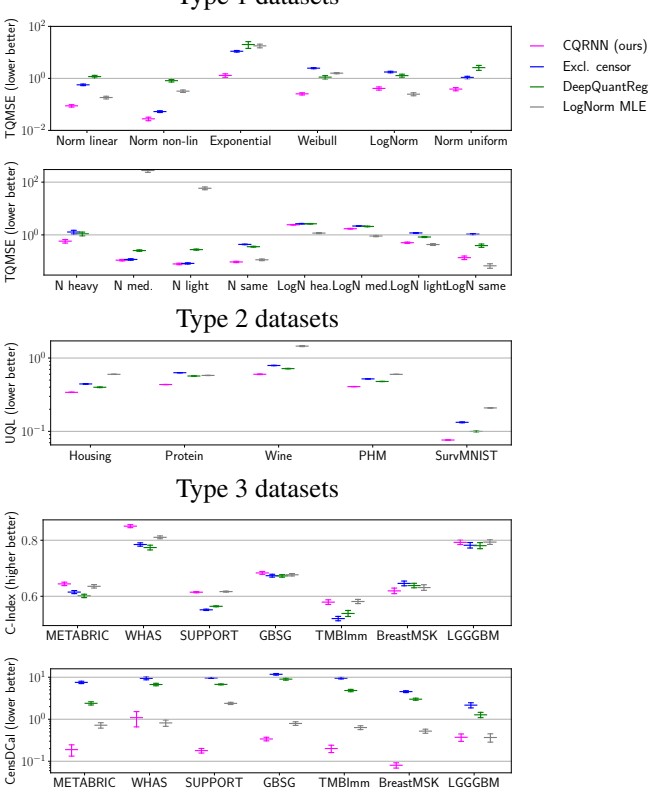

Figure 2: Main benchmarking results. Mean $\pm$ one standard error over ten runs.

## 6.3 Comparison of Sequential Grid and CQRNN

We ran a head-to-head comparison of CQRNN and the sequential grid algorithm on all type 1 synthetic datasets, and all type 3 real datasets. Table 1 compares on type 1 datasets, along with statistical tests of significance for TQMSE differences (200 random seeds). Appendix Table 5 provides results for the type 3 datsets, reporting C-index and CensDCal (50 random seeds). Significance tests are described in Appendix B.1.

CQRNN delivers benefits in speed up both at training and test time, running an order of magnitude faster than the sequential grid algorithm. CQRNN also dramatically reduces model size. The magnitude of these benefits is largely determined by the size of the quantile grid, $M$, which is explicit in the sequential algorithm's time and space complexity, $\mathcal{O}(M)$, but largely avoided in CQRNN.

Differences in the quality of quantiles of the two algorithms is usually slight (after-all both leverage the same estimator), though CQRNN shows statistically significant gains on 8/14 type 1 datasets in terms of TQMSE, 4/7 type 3 datsets in terms of C-Index, and 7/7 type 3 datasets in terms of CensDCal. We hypothesise that having a single NN output all quantiles provides a helpful inductive bias, encouraging similarity between adjacent quantiles, that's not present in independently trained NNs.

## 6.4 Hyperparameter Investigation

The CQRNN algorithm includes two hyperparameters – the grid of quantiles, $\mathrm{grid}_\tau$, and the pseudo y value, $y^*$. We ran ablations to empirically investigate the effect of these, as well as remedies for the 'crossing-quantile' problem, and a comparison of partial vs. full optimisation. Here we summarise our findings, Appendix C.1 provides full details.

We tested grid sizes, $M \in \{9, 19, 39\}$, on several type 1 datasets. In general, a finer grid (larger $M$) is slightly beneficial, though there is variance between datasets and gains are sometimes only seen

for larger datasets (>5,000 datapoints). Selection of $y^*$ requires some care for CQRNN. We defined it in terms of the maximum $y$ value in the training set, $y^* = c_{y^*} \max_i y_i$, for a hyperparemeter, $c_{y^*} > 1$. Performance can be improved by tuning $c_{y^*}$ for each dataset, but using $c_{y^*} = 1.2$ provided consistently reasonable results – this was the value used in our benchmarking experiments. We trialled two methods for combating the crossing-quantile problem. 1) Adding a crossing loss penalty. 2) Constraining the NN architecture to output quantiles adding to the previous prediction. Neither of these methods significantly affected performance.

## 7   Discussion & Conclusion

This paper has taken a popular idea from survival analysis, Portnoy's censored quantile regression estimator, and shown how it can be efficiently combined with NNs in a new algorithm, CQRNN. We provided theoretical insight by interpreting it as a flavour of EM. Empirically the method outperformed existing approaches, consistently producing more accurate quantile estimates across a range of synthetic and real datasets. For example, across datasets of type 2 and 3, CQRNN was best calibrated in 10 out of 12 instances (see CensDCal in Table 4).

**Limitations.** Firstly, our theoretical results have not said that solutions will converge on a global optimum. Secondly, we have only tested CQRNN on a modest number of real-world datasets, many drawn from the biomedical domain. It's possible that some datasets may cause CQRNN problems, as we found on BreastMSK (Figure 2). In particular, cases where higher quantiles are undefined should be handled with care.

**Conclusion.** To summarise, our work contributes toward unlocking the benefit that modern machine learning could bring to important domains such as healthcare and machinery prognostics. By outputting quantiles, CQRNN naturally communicates a measure of uncertainty in its predictions, which makes it particularly suitable to these high-stakes applications, and a valuable addition to the toolkit combining deep NNs with survival analysis.

Table 2: Summary of all datasets used.

| Dataset | Feats | Train data | Test data | Prop. censored | Target sampling distribution | Censoring sampling distribution |
|---|---|---|---|---|---|---|
| **Type 1 – Synthetic datasets with synthetic censoring** | | | | | | |
| Norm linear | 1 | 500 | 1000 | 0.20 | $\mathcal{N}(2x + 10, (x + 1)^2)$ | $\mathcal{N}(4x + 10, (0.8x + 0.4)^2)$ |
| Norm non-linear | 1 | 500 | 1000 | 0.24 | $\mathcal{N}(x\sin(2x) + 10, (0.5x + 0.5)^2)$ | $\mathcal{N}(2x + 10, 2^2)$ |
| Exponential | 1 | 500 | 1000 | 0.30 | $\text{Exp}(2x + 4)$ | $\text{Exp}(-3x + 15)$ |
| Weibull | 1 | 500 | 1000 | 0.22 | $\text{Weibull}(4x\sin(2(x - 1)) + 10, 5)$ | $\text{Weibull}(-3x + 20, 5)$ |
| LogNorm | 1 | 500 | 1000 | 0.21 | $\text{Lognorm}((x - 1)^2, x^2)$ | $\mathcal{U}(0, 10)$ |
| Norm uniform | 1 | 500 | 1000 | 0.62 | $\mathcal{N}(2x\cos(2x) + 13, (x^2 + 0.5)^2)$ | $\mathcal{U}(0, 18)$ |
| Norm heavy | 4 | 2000 | 1000 | 0.80 | $\mathcal{N}(3x_0 + x_1^2 - x_2^2 + 2\sin(x_2 x_3) + 6, (x^2 + 0.5)^2)$ | $\mathcal{U}(0, 12)$ |
| Norm med. | 4 | 2000 | 1000 | 0.49 | — " — | $\mathcal{U}(0, 20)$ |
| Norm light | 4 | 2000 | 1000 | 0.25 | — " — | $\mathcal{U}(0, 40)$ |
| Norm same | 4 | 2000 | 1000 | 0.50 | — " — | Equal to target dist. |
| LogNorm heavy | 8 | 4000 | 1000 | 0.75 | $\text{Lognorm}(\sum_i^8 \beta_i x_i, 1)/10$ | $\mathcal{U}(0, 0.4)$ |
| LogNorm med. | 8 | 4000 | 1000 | 0.52 | — " — | $\mathcal{U}(0, 1.0)$ |
| LogNorm light | 8 | 4000 | 1000 | 0.23 | — " — | $\mathcal{U}(0, 3.5)$ |
| LogNorm same | 8 | 4000 | 1000 | 0.50 | — " — | Equal to target dist. |
| **Type 2 – Real datasets with synthetic censoring** | | | | | | |
| Housing | 8 | 16512 | 4128 | 0.50 | Real | $\mathcal{U}(0, c_1)$ |
| Protein | 9 | 36584 | 9146 | 0.44 | Real | $\mathcal{U}(0, c_2)$ |
| Wine | 11 | 5197 | 1300 | 0.69 | Real | $\mathcal{U}(0, c_3)$ |
| PHM | 21 | 36734 | 9184 | 0.52 | Real | $\mathcal{U}(0, c_4)$ |
| SurvMNIST | $28\times28$ | 48000 | 12000 | 0.53 | One Gamma dist. per MNIST class | $\mathcal{U}(0, c_5)$ |
| **Type 3 – Real datasets with real censoring** | | | | | | |
| METABRIC | 9 | 1523 | 381 | 0.42 | Real | Real |
| WHAS | 6 | 1310 | 328 | 0.57 | Real | Real |
| SUPPORT | 14 | 7098 | 1775 | 0.32 | Real | Real |
| GBSG | 7 | 1785 | 447 | 0.42 | Real | Real |
| TMBImmuno | 3 | 1328 | 332 | 0.49 | Real | Real |
| BreastMSK | 5 | 1467 | 367 | 0.77 | Real | Real |
| LGGGBM | 5 | 510 | 128 | 0.60 | Real | Real |

## Acknowledgement

This work was supported by the National Key Research and Development Program of China (2020AAA0106302); NSF of China Projects (Nos. 62061136001, U19B2034, U1811461, U19A2081); Beijing NSF Project (No. JQ19016); a grant from Tsinghua Institute for Guo Qiang; and the High Performance Computing Center, Tsinghua University. J.Z was also supported by the XPlorer Prize.

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
