# A  Analytical Results

## A.1  Proofs

**Theorem 1.** *Let the likelihood for each datapoint at each quantile be an asymmetric Laplace distribution with scale, $\lambda = \sqrt{\tau - \tau^2}$, and asymmetry, $k = \tau/\sqrt{\tau - \tau^2}$. The negative log likelihood is,*

$$- \log p(y|\mathbf{x}, \theta, \mathbf{w}, y^*) = \sum_{\tau \in grid_\tau} \mathcal{L}_{Port.}(\theta, y, \mathbf{x}, \tau, \mathbf{w}, y^*) + \text{constant} . \tag{11}$$

*Proof.* Define the likelihood over all quantiles of interest, and split censored datapoints into two pseudo datapoints, one at the censoring location and one at the large pseudo value $y^*$, to give a weighted likelihood,

$$p(y|\mathbf{x}, \theta, \mathbf{w}, y^*) = \prod_{\tau \in grid_\tau} p(y|\mathbf{x}, \theta, \mathbf{w}, y^*, \tau), \tag{12}$$

$$p(y|\mathbf{x}, \theta, \mathbf{w}, y^*, \tau) = \prod_{i \in \mathcal{S}_{observed}} p(y_i|\mathbf{x}_i, \theta) \prod_{j \in \mathcal{S}_{censored}} p(y_j|\mathbf{x}_j, \theta)^{w_j} p(y^*|\mathbf{x}_j, \theta)^{1-w_j}. \tag{13}$$

We write the asymmetric Laplace density with $\hat{y}_{j,\tau}$ as the location parameter and scale $\lambda$ and asymmetry $k$,

$$f(y_j; \hat{y}_{j,\tau}, \lambda, k) = \frac{\lambda}{k + 1/k} \begin{cases} \exp((\lambda/k)(y_j - \hat{y}_{j,\tau})) & \text{if } \hat{y}_{j,\tau} > y_j \\ \exp(-\lambda k(y_j - \hat{y}_{j,\tau})) & \text{else} \end{cases} . \tag{14}$$

Setting $\lambda = \sqrt{\tau - \tau^2}$ and $k = \tau/\sqrt{\tau - \tau^2}$ and rearranging,

$$f(y_j; \hat{y}_{j,\tau}, \lambda, k) = (\tau - \tau^2) \exp(y_j - \hat{y}_{j,\tau})(-\tau + \mathbb{I}[\hat{y}_{j,\tau} > y_j]) \tag{15}$$

$$\log f(y_j; \hat{y}_{j,\tau}, \lambda, k) = -\rho_\tau(y_j, \hat{y}_{j,\tau}) + \text{constant} \tag{16}$$

Taking the logarithm of Eq. 13 we have,

$$\log p(y|\mathbf{x}, \theta, \mathbf{w}, y^*) =$$
$$\sum_{\tau \in grid_\tau} \sum_{i \in \mathcal{S}_{observed}} \log p(y_i|x_i, \theta) + \sum_{j \in \mathcal{S}_{censored}} \log p(y_j|x_j, \theta)^{w_j} + \log p(y^*|x_j, \theta)^{1-w_j}. \tag{17}$$

Eq. 16 may be substituted into this if all likelihoods of observed and censored pseudo datapoints are chosen to follow asymmetric Laplace distributions. Taking the negative then recovers the theorem's result.

$\square$

**Theorem 2.** *If $\hat{q}_j$ is underestimated, one iteration of the algorithm acts to increase the quantile predictions, $\hat{y}_{j,\tau}$, by the same amount, or even higher, than if the weight had been correct. If $\hat{q}_j$ is overestimated, $\hat{y}_{j,\tau}$, is decreased by the same amount, or even lower, than with the correct weight.*

**Remark.** *Note that if a quantile $\hat{q}_j$ is underestimated, it's desirable to increase the quantiles relating to that datapoint, for input $\mathbf{x}_j$ and predictions $\hat{y}_{j,\tau}$. If a quantile $\hat{q}_j$ is instead overestimated, it's desirable to decrease the quantiles for input $\mathbf{x}_j$ and predictions $\hat{y}_{j,\tau}$. We refer to this desired behaviour as 'self-correcting'.*

*Proof.* Denote $q_j$ the true quantile that censored datapoint $j$ is censored in, and $w_j = (\tau - q_j)/(1 - q_j)$ the corresponding true weight. We now consider one iteration of the algorithm in the case that the estimated quantile is underestimated.

1. Model underpredicts the censored quantile, $\hat{q}_j = q_j - \epsilon$, for some $\epsilon > 0$.

2. The corresponding weight is also underestimated, $\hat{q}_j < q_j \implies \hat{w}_j < w_j$, shown in lemma 2.1.

3. Lemma 2.3 shows that, for censored datapoint $j$, the gradient of Eq. 4 wrt the quantile prediction $\hat{y}_{j,\tau}$ is,

$$\frac{\partial \mathcal{L}_{\text{Port.}}(\theta, \mathcal{D}, \tau, \mathbf{w}, y^*)}{\partial \hat{y}_{j,\tau}} = \begin{cases} -\tau & \text{if, } \hat{y}_{j,\tau} < y_j \\ w_j - \tau & \text{if, } y_j \leq \hat{y}_{j,\tau} < y^* \\ 1 - \tau & \text{if, } y^* \leq \hat{y}_{j,\tau} \end{cases} . \tag{18}$$

Hence, if $\hat{w}_j$ is underestimated it holds that, $\frac{\partial \mathcal{L}_{\text{Port.}}(\theta, \mathcal{D}, \tau, \hat{\mathbf{w}}, y^*)}{\partial \hat{y}_{j,\tau}} \leq \frac{\partial \mathcal{L}_{\text{Port.}}(\theta, \mathcal{D}, \tau, \mathbf{w}, y^*)}{\partial \hat{y}_{j,\tau}}$.

4. When this gradient is used for optimisation, this has the effect of increasing the quantile prediction, $\hat{y}_{j,\tau}$, by either the same amount, or even higher, than if the weight had been correct.

Similar (reversed) logic applies if the quantile is initially overestimated, $\hat{q}_j = q_j + \epsilon$, (e.g. lemma 2.2) which encourages decreasing the quantile predictions, $\hat{y}_{j,\tau}$, by the same amount or lower than with correct weights. Hence, weight estimates will be improved in future iterations of the algorithm. Note that this applies to all quantiles, $\tau \in \text{grid}_\tau$.

$\square$

**Lemma 2.1.** *Let, $\hat{q}_j = q_j - \epsilon$, and $\epsilon > 0$. It holds that, $\hat{q}_j < q_j \implies \hat{w}_j < w_j$, for, $\tau \in (0, 1)$, and, $\hat{q}_j, q_j \in (0, 1)$.*

*Proof.* From the definition of the weights, $\hat{w}_j < w_j \implies (\tau - q_j - \epsilon)/(1 - q_j - \epsilon) < (\tau - q_j)/(1 - q_j)$. Let $a := \tau - q_j$ and $b := 1 - q_j$. Note that, $a < b$, since by assumption, $\tau < 1$. We must show that,

$$\frac{a - \epsilon}{b - \epsilon} < \frac{a}{b} \tag{19}$$

$$\frac{a - \epsilon}{b - \epsilon} \frac{b}{a} < 1 \tag{20}$$

$$\frac{ab - b\epsilon}{ab - a\epsilon} < 1, \text{ which holds since, } a < b. \tag{21}$$

$\square$

**Lemma 2.2.** *Let, $\hat{q}_j = q_j + \epsilon$, and $\epsilon > 0$. It holds that, $\hat{q}_j > q_j \implies \hat{w}_j > w_j$, for, $\tau \in (0, 1)$, and, $\hat{q}_j, q_j \in (0, 1)$.*

*Proof.* This proof follows lemma 2.1. We now have,

$$\frac{ab + b\epsilon}{ab + a\epsilon} > 1, \text{ which holds since, } a < b. \tag{22}$$

$\square$

**Lemma 2.3.** *The partial derivative of Portnoy's loss wrt the predicted quantile, for censored datapoint $j$, is given by,*

$$\frac{\partial \mathcal{L}_{Port.}(\theta, \mathcal{D}, \tau, \mathbf{w}, y^*)}{\partial \hat{y}_{j,\tau}} = \begin{cases} -\tau & \text{if, } \hat{y}_{j,\tau} < y_j \\ w_j - \tau & \text{if, } y_j \leq \hat{y}_{j,\tau} < y^* \\ 1 - \tau & \text{if, } y^* \leq \hat{y}_{j,\tau} \end{cases} . \tag{23}$$

*Proof.* Recalling that,

$$\rho_\tau(y_i, \hat{y}_{i,\tau}) = (y_i - \hat{y}_{i,\tau})(\tau - \mathbb{I}[\hat{y}_{i,\tau} > y_i]), \tag{24}$$

we have,

$$\frac{\partial \rho_\tau(y_i, \hat{y}_{i,\tau})}{\partial \hat{y}_{j,\tau}} = \mathbb{I}[\hat{y}_{i,\tau} > y_i] - \tau. \tag{25}$$

We are interested in the derivative for the censored portion of Portnoy's loss in Eq. 4,

$$\frac{\partial w_j \rho_\tau(y_j, \hat{y}_{j,\tau}) + (1 - w_j)\rho_\tau(y^*, \hat{y}_{j,\tau})}{\partial \hat{y}_{j,\tau}}. \tag{26}$$

Note that we always choose, $y^* > y_j$. Hence $\hat{y}_{j,\tau} < y_j \implies \hat{y}_{j,\tau} < y^*$, so there are three cases to consider. Case 1, $\hat{y}_{j,\tau} < y_j$, case 2, $y_j \le \hat{y}_{j,\tau} < y^*$, case 3 $y^* \le \hat{y}_{j,\tau}$.

For case 1,

$$\frac{\partial w_j \rho_\tau(y_j, \hat{y}_{j,\tau}) + (1 - w_j)\rho_\tau(y^*, \hat{y}_{j,\tau})}{\partial \hat{y}_{j,\tau}} = w_j(-\tau) + (1 - w_j)(-\tau) = -\tau. \tag{27}$$

For case 2,

$$\frac{\partial w_j \rho_\tau(y_j, \hat{y}_{j,\tau}) + (1 - w_j)\rho_\tau(y^*, \hat{y}_{j,\tau})}{\partial \hat{y}_{j,\tau}} = w_j(1 - \tau) + (1 - w_j)(-\tau) = w_j - \tau. \tag{28}$$

For case 3,

$$\frac{\partial w_j \rho_\tau(y_j, \hat{y}_{j,\tau}) + (1 - w_j)\rho_\tau(y^*, \hat{y}_{j,\tau})}{\partial \hat{y}_{j,\tau}} = w_j(1 - \tau) + (1 - w_j)(1 - \tau) = 1 - \tau. \tag{29}$$

$\square$

## A.2 First Iteration of the Sequential Grid Algorithm

In this section we compare the procedure proposed by Portnoy [2003] to find the first quantile predicted at, $\tau_0 := \mathrm{grid}_\tau[0]$, with the procedure we propose in the sequential grid algorithm for NNs (Algorithm 1). We show that these two procedures produce equivalent gradients.

**Portnoy's procedure.** Portnoy [2003] require that no censored datapoints lie below the first quantile, and propose deleting these from the dataset when this does occur, as follows.

1. Set $\mathbf{w} = \mathbf{1}$ for all censored datapoints in dataset.
2. Optimise $\mathcal{L}_{\mathrm{Port.}}$ (Eq. 4) for $\tau_0$.
3. Find all censored datapoints below $\tau_0$, $B \leftarrow \{j \in \mathcal{S}_{\mathrm{censored}} : y_j < \hat{y}_{j,\tau_0}\}$.
4. If $B$ is empty then exit.
5. Exclude all elements of $B$ from the dataset and repeat.

This optimisation could be repeated many times. We'd like to avoid this since training NNs on potentially large datasets can be costly.

**Sequential grid for NNs procedure.** Algorithm 1 instead simply sets $\mathbf{q} = \mathbf{0}$, and optimises the first quantile once only.

1. Set $\mathbf{q} = \mathbf{0}$ for all censored datapoints in dataset.
2. Optimise $\mathcal{L}_{\mathrm{Port.}}$ (Eq. 4) for $\tau_0$.

**Justification.** We now justify why this is a reasonable approximation. First note that when $\hat{q}_j = 0$ we have, $\hat{w}_i = \frac{\tau - \hat{q}_i}{1 - \hat{q}_i} = \tau$. Using lemma 2.3 we can compare the gradients for each procedure.

For Portoy's procedure, when $\hat{w}_j = 1$,

$$\frac{\partial \mathcal{L}_{\mathrm{Port.}}(\theta, \mathcal{D}, \tau, \hat{\mathbf{w}}, y^*)}{\partial \hat{y}_{j,\tau}} = \begin{cases} -\tau & \text{if, } \hat{y}_{j,\tau} < y_j \\ 1 - \tau & \text{if, } y_j \le \hat{y}_{j,\tau} < y^* \implies \text{ set to 0 in next iteration .} \\ 1 - \tau & \text{if, } y^* \le \hat{y}_{j,\tau} \end{cases} \tag{30}$$

For Algorithm 1, when $\hat{q}_j = 0$,

$$\frac{\partial \mathcal{L}_{\mathrm{Port.}}(\theta, \mathcal{D}, \tau, \hat{\mathbf{w}}, y^*)}{\partial \hat{y}_{j,\tau}} = \begin{cases} -\tau & \text{if, } \hat{y}_{j,\tau} < y_j \\ 0 & \text{if, } y_j \le \hat{y}_{j,\tau} < y^* \\ 1 - \tau & \text{if, } y^* \le \hat{y}_{j,\tau} \end{cases} \tag{31}$$

At first look, the gradients appear to differ in the case $y_j \le \hat{y}_{j,\tau} < y^*$. But when a datapoint triggers this criteria in Portnoy's procedure, it will be excluded and the model retrained, in which case its gradient becomes 0. As such the gradients for a censored datapoint in both procedures are equivalent.

# B  Experimental Details

This section provides further details about all experiments run. Our code base uses the $\mathrm{PyTorch}$ framework. Hyperparameters in Appendix B.1. Metrics in Appendix B.2. Baselines in Appendix B.3. Datasets in Appendix B.4.

**Hardware.** We used an internal cluster for experiments, utilising machines with four GPUs and 14 CPU cores. Most of our datasets used fully-connected NNs, which were trained on CPU, while GPUs were used for the SurvMNIST experiments.

## B.1  Full Hyperparameter Details

Below we list hyperparameter settings used and where applicable the tuning protocols followed.

### B.1.1  Qualitative 1D Analysis

Section 6.1 experiment. All methods used the same optimisation procedure and NN architecture, without tuning.

- Training dataset size: 500, where, $\mathbf{x} \sim \mathcal{U}(0,2)$
- Epochs: 100
- Optimiser: Adam
- Learning rate: 0.01 (decreased 70% and 90% of the way through training)
- Batch size: 128
- Weight decay: 0.0001
- NN architecture: Fully-connected, two hidden layers of 100 hidden nodes, GeLU activations
- $y^* = 1.2 \times \max_i y_i$
- $\mathrm{grid}_\tau \in \{0.1, 0.3, 0.5, 0.7, 0.9\}$

### B.1.2  Benchmarking

Section 6.2 experiment. Experiments were repeated over 10 random seeds. Hyperparameter settings.

- Training dataset size: various – see Table 2
- Test dataset size: various – see Table 2
- Epochs: $\in \{10, 20, 50, 100\}$
- Optimiser: Adam
- Learning rate: 0.01 for fc NN, 0.001 for CNN (decreased 70% and 90% of the way through training)
- Batch size: 128
- Weight decay: 0.0001
- Default NN architecture: Fully-connected, two hidden layers of 100 hidden nodes, ReLU activations
- CNN architecture for SurvMNIST following Goldstein et al. [2020]: Conv2D[64, (5×5)] → ReLU → Dropout(0.2) → AveragePool(2×2) → Conv2D[128, (5×5)] → ReLU → Dropout(0.2) → AveragePool(2×2) → Conv2D[256, (2×2)] → ReLU → Linear
- $y^* = 1.2 \times \max_i y_i$
- Grid size $M \in \{6, 10, 20\}$
- Dropout $\in \{\mathrm{True}, \mathrm{False}\}$

Tuning process for epochs and dropout:

- For the real type 2 and type 3 datasets, we tuned number of epochs $\in \{10, 20, 50, 100\}$ and dropout $\in \{\mathrm{True}, \mathrm{False}\}$ for each dataset for each method. We used three random splits as a validation (but not overlapping with the random seeds used in the final test run).

- Epochs was fixed to 100 and dropout disabled for: Norm linear, Norm non-linear, Exponential, Weibull, LogNorm, Norm uniform.

- Epochs was fixed to 20 and dropout disabled for: Norm heavy, Norm medium, Norm light, Norm same.

Table 3: Availability of metrics for each dataset type.

| Dataset type | Target distribution | Censoring distribution | TQMSE | UQL | UnDCal | CensDCal | C-Index |
|---|---|---|---|---|---|---|---|
| Type 1 | Synthetic | Synthetic | ✓ | ✓ | ✓ | ✓ | ✓ |
| Type 2 | Real | Synthetic | ✗ | ✓ | ✓ | ✓ | ✓ |
| Type 3 | Real | Real | ✗ | ✗ | ✗ | ✓ | ✓ |

- Epochs was fixed to 10 and dropout disabled for: LogNorm heavy, LogNorm medium, LogNorm light, LogNorm same.

We fixed the grid size according to an estimate of how densely the datapoints covered the input space (a rough consideration of dataset size and number of features):

- Grid size $M = 5$ for smaller datasets with more features: WHAS, SUPPORT, GBSG, TMBImmuno, BreastMSK, LGGGBM, METABRIC.

- Grid size $M = 9$ for medium datasets or smaller datasets with less features: Norm linear, Norm non-linear, Exponential, Weibull, LogNorm, Norm uniform.

- Grid size $M = 19$ for larger datasets or those with less features: Norm heavy, Norm medium, Norm light, Norm same, LogNorm heavy, LogNorm medium, LogNorm light, LogNorm same, Housing, Protein, Wine, PHM, SurvMNIST.

### B.1.3   Comparison of Sequential Grid and CQRNN

Section 6.3 experiment. This was carried out under the same protocol as for the main benchmarking, but repeated over a larger number of seeds (200 for type 1 datasets, 50 for type 3 datasets). To obtain 95% confidence intervals, we compute the standard error of the difference between means, and multiply it by a two-sided t-statistic, with (number of seeds$-1$) degrees of freedom at the $\alpha = 0.05$ significance level. If zero falls within this confidence interval, the difference is deemed not significant.

### B.1.4   Hyperparameter Investigation

Section 6.4 experiment. Hyperparameters are as for the main benchmarking except $M$ and $y^*$ were varied as stated in the text. Epochs were set via the formula epochs $= 500 \times 200/N$ ensuring the same number of gradient updates were made on each run. Experiments were repeated over 100 random seeds for the grid investigation, and ten random seeds for the $y^*$ investigation.

### B.2   Metrics

This section briefly expands upon the metrics introduced in Section 6. Table 3 summarises the availability on each metric for each dataset type. The computation of each is also detailed below.

$$\text{True quantile MSE (TQMSE)} := \frac{1}{N} \sum_{\tau \in [0.1, 0.5, 0.9]} \sum_{i=1}^{N} (\hat{y}_{i,\tau} - y_{i,\tau})^2 \tag{32}$$

$$\text{Uncensored quantile loss (UQL)} := \frac{1}{N} \sum_{\tau \in [0.1, 0.5, 0.9]} \sum_{i=1}^{N} \rho_\tau(y_i, \hat{y}_{i,\tau}) \tag{33}$$

$$\text{Uncensored D-Calibration (UnDCal)} := 100 \times \sum_{j=1}^{M-1} \left( (\tau_{j+1} - \tau_j) - \frac{1}{N} \sum_{i=1}^{N} \mathbb{I}[\hat{y}_{i,\tau_j} < y_i \le \hat{y}_{i,\tau_{j+1}}] \right)^2 \tag{34}$$

$$\text{Censored D-Calibration (CensDCal)} := 100 \times \sum_{j=1}^{M-1} \left( (\tau_{j+1} - \tau_j) - \frac{1}{N} \xi \right)^2 \tag{35}$$

where, Goldstein et al. [2020] defines,

$$\xi = \sum_{i \in \mathcal{S}_{\text{observed}}} \mathbb{I}[\hat{y}_{i,\tau_j} < y_i \leq \hat{y}_{i,\tau_{j+1}}] + \sum_{i \in \mathcal{S}_{\text{censored}}} \frac{(\tau_{j+1} - q_i)\mathbb{I}[\hat{y}_{i,\tau_j} < y_i \leq \hat{y}_{i,\tau_{j+1}}]}{1 - q_i} + \frac{(\tau_{j+1} - \tau_j)\mathbb{I}[q_i < \tau_j]}{1 - q_i}.$$

(36)

We increase the magnitude of DCal metrics by $100\times$ to make the numbers of similar order to TQMSE and UQL.

### B.3 Baselines

This section provides some extra detail about the LogNorm MLE baseline. For this method, we use a NN with two outputs, representing the mean, $\mu$, and standard deviation, $\sigma$, of a Log Normal distribution, i.e. $\log y \sim \mathcal{N}(\hat{\mu}, \hat{\sigma}^2)$. We pass the output representing the standard deviation prediction through a $\mathrm{SoftPlus}$ to ensure it is always positive and differentiable.

The maximum likelihood estimation loss is then,

$$-\mathcal{L}_{\text{MLE}}(\theta, \mathcal{D}) \coloneqq \sum_{i \in \mathcal{S}_{\text{observed}}} \log p(y_i | \mathbf{x}_i, \theta) + \sum_{j \in \mathcal{S}_{\text{censored}}} \log(1 - \mathrm{CDF}(y_j | \mathbf{x}_j, \theta)),$$

(37)

where the likelihood and CDF follow the analytical expressions for the Log Normal distribution. At evaluation time, we use the $\mathrm{SciPy}$ package to compute the quantiles from the predicted Log Normal distribution that correspond to those in $\mathrm{grid}_\tau$. This allows a like-for-like comparison with our other baselines.

### B.4 Dataset Details

This section provides further detail about the source of each dataset used. All real-world datasets were taken from open-access repositories, and had already been anonymised. For real-world datasets, we do not follow any previous test/train splits, rather we randomly shuffle the data for each run, selecting 80% for training and 20% for testing.

#### B.4.1 Type 1 Datasets

Table 2 details the generating target and censoring distributions used, as well as numbers of test and train datapoints. Inputs were always generated uniformly via, $\mathbf{x} \sim \mathcal{U}(0,2)^D$ for $D$ features.

#### B.4.2 Type 2 Datasets

Table 2 details dataset sizes and number of features. All datasets used a uniform censoring distribution, $c_i \sim \mathcal{U}(0,c)$, where $c$ was selected to be equal to the 90th percentile of the target distribution for SurvMNIST, and $c = 1.5 \max_i y_i$ for the other type 2 datasets. Housing was sourced from $\mathtt{Scikit\ Learn}$ datasets, while Protein, Wine and PHM were sourced through OpenML https://www.openml.org/.

- **Housing** Target is median house prices. Retrieved from https://scikit-learn.org/stable/datasets/real_world.html#california-housing-dataset.
- **Protein** Target is RMSD. OpenML lookup ID is $\mathtt{physicochemical\text{-}protein}$.
- **Wine** Target is quality of wine. OpenML lookup ID is $\mathtt{wine\ quality}$.
- **PHM** (prognostics health management) target is breakdown time of simulated machines. OpenML lookup ID is $\mathtt{NASA\ PHM2008}$.

Our final type 2 dataset is slightly different, since we don't use the provided labels directly.

- **SurvMNIST** appeared in Goldstein et al. [2020], who adapted it from Sebastian Pölsterl's blog: https://k-d-w.org/blog/2019/07/survival-analysis-for-deep-learning/. It uses the standard MNIST dataset http://yann.lecun.com/exdb/mnist/, but targets are drawn from a Gamma distribution, with different parameters per class. Goldstein et al. [2020] used a small variance fixed across classes, with means $\in [11.25, 2.25, 5.25, 5.0, 4.75, 8.0, 2.0, 11.0, 1.75, 10.75]$. For our purposes, we are most interested in how well methods capture variance, so we vary it, variance $\in [0.1, 0.5, 0.1, 0.2, 0.2, 0.2, 0.3, 0.1, 0.4, 0.6]$.

### B.4.3 Type 3 Datasets

We provide a brief overview of each dataset. Four datasets – GBSG, METABRIC, SUPPORT, WHAS – were all retrieved from https://github.com/jaredleekatzman/DeepSurv/tree/master/experiments/data. Katzman et al. [2018] provides a detailed introduction to these datasets. The other three datasets – TMBImmnuo, BreastMSK, LGGGBM – were all sourced from the cBioPortal, https://www.cbioportal.org/, for cancer genomics.

- **GBSG** (Rotterdam & German Breast Cancer Study Group) requires prediction of survival time for breast cancer patients.

- **METABRIC** (Molecular Taxonomy of Breast Cancer International Consortium) requires prediction of survival time for breast cancer patients. Covariates include expressions for four genes as well as clinical data.

- **SUPPORT** (Study to Understand Prognoses Preferences Outcomes and Risks of Treatment) requires prediction of survival time in seriously ill hospitalised patients. Covariates include demographic and basic diagnosis information.

- **WHAS** (Worcester Heart Attack Study) requires prediction of acute myocardial infraction survival.

- **TMBImmuno** (Tumor Mutational Burden and Immunotherapy) requires prediction of survival time for patients with various cancer types using clinical data. Covariates include age, sex, and number of mutations. Retrieved from https://www.cbioportal.org/study/clinicalData?id=tmb_mskcc_2018.

- **BreakMSK** requires prediction of survival time for patients with breast cancer using tumour information. Covariates include ER, HER, HR, mutation count, TMB. Retrieved from https://www.cbioportal.org/study/clinicalData?id=breast_msk_2018.

- **LGGGBM.** requires prediction of survival time for cancer patient from clinical data. Covariates include age, sex, purity, mutation count, TMB. Retrieved from https://www.cbioportal.org/study/clinicalData?id=lgggbm_tcga_pub

# C   Further Results

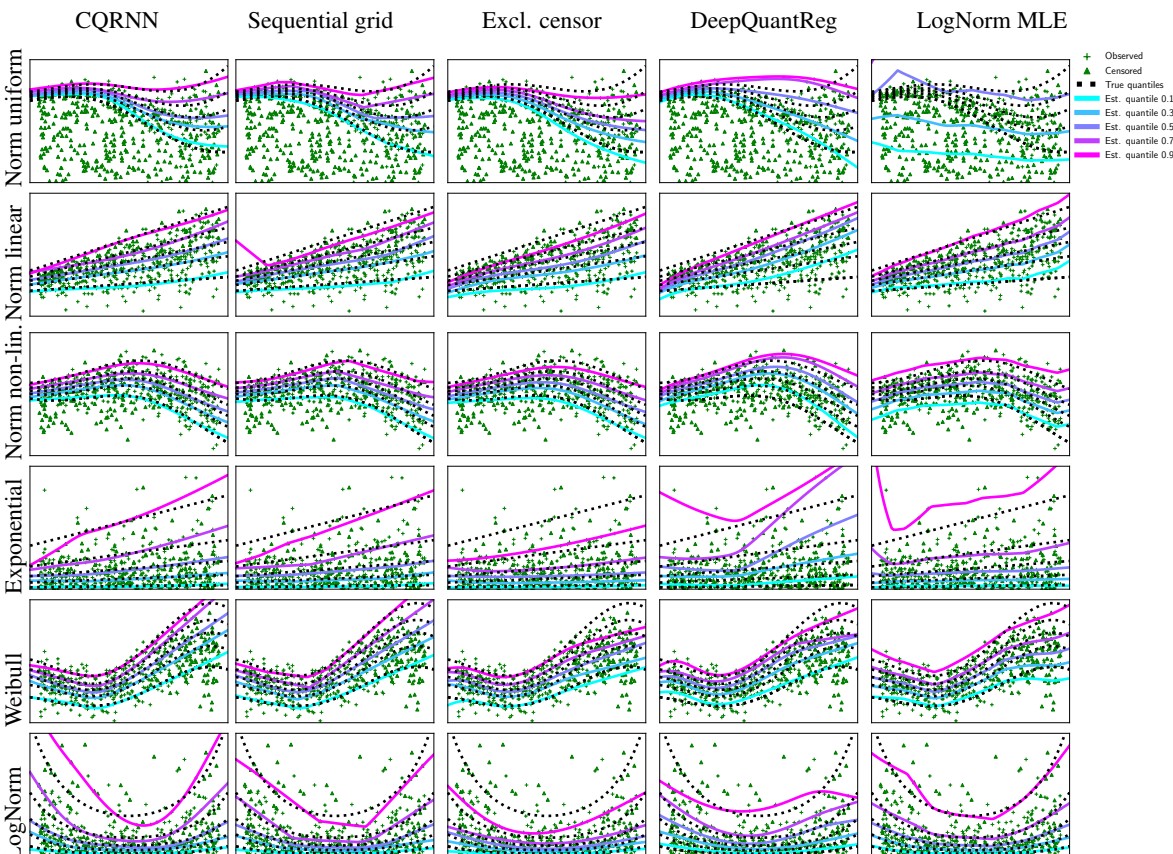

Figure 3: This figure shows estimated quantiles (blue through pink) compared to ground truth quantiles (dashed black lines). It shows 1D synthetic datasets of varying functions and noise distributions (rows), fitted by various methods (columns).

Table 4: Full results table for all datasets, methods and metrics. Mean ± 1 standard error for test set over 10 runs.

| Dataset | Method | MSE to true quantile (lower better) | Uncensored quantile loss (lower better) | Uncensored D-Calibration (lower better) | Concordance-Index (higher better) | Censored D-Calibration (lower better) |
|---|---|---|---|---|---|---|
| | | **Type 1 – synthetic data, synthetic censoring** | | | | |
| Norm linear | CQRNN | **0.088 ± 0.009** | **1.517 ±0.01** | 0.327 ±0.057 | **0.663 ±0.002** | 0.211 ±0.034 |
| Norm linear | Sequential grid | 2.779 ± 0.506 | 1.623 ±0.02 | **0.303 ±0.06** | 0.662 ±0.002 | 0.224 ±0.024 |
| Norm linear | Excl. censor | 0.566 ± 0.039 | 1.62 ±0.016 | 3.078 ±0.244 | 0.662 ±0.002 | 2.349 ±0.102 |
| Norm linear | DeepQuantReg | 1.172 ± 0.133 | 1.695 ±0.026 | 1.94 ±0.192 | **0.663 ±0.002** | 1.241 ±0.138 |
| Norm linear | LogNorm MLE | 0.184 ± 0.021 | 1.522 ±0.008 | 0.315 ±0.023 | 0.662 ±0.002 | 0.232 ±0.04 |
| Norm non-lin | CQRNN | 0.028 ± 0.004 | **0.759 ±0.005** | 0.262 ±0.044 | **0.674 ±0.004** | **0.186 ±0.023** |
| Norm non-lin | Sequential grid | **0.027 ± 0.003** | **0.759 ±0.005** | 0.283 ±0.034 | 0.673 ±0.004 | 0.194 ±0.03 |
| Norm non-lin | Excl. censor | 0.053 ± 0.004 | 0.768 ±0.006 | 0.579 ±0.088 | 0.673 ±0.004 | 0.449 ±0.054 |
| Norm non-lin | DeepQuantReg | 0.818 ± 0.093 | 0.998 ±0.019 | 4.263 ±0.234 | 0.609 ±0.023 | 2.593 ±0.206 |
| Norm non-lin | LogNorm MLE | 0.323 ± 0.035 | 0.824 ±0.009 | 1.651 ±0.118 | 0.653 ±0.005 | 0.971 ±0.077 |
| Exponential | CQRNN | **1.298 ± 0.201** | **4.057 ±0.04** | 0.404 ±0.035 | **0.559 ±0.003** | 0.248 ±0.031 |
| Exponential | Sequential grid | 1.702 ± 0.397 | 4.061 ±0.039 | **0.391 ±0.057** | 0.558 ±0.004 | **0.226 ±0.025** |
| Exponential | Excl. censor | 11.03 ± 0.772 | 4.456 ±0.039 | 2.537 ±0.18 | 0.535 ±0.007 | 1.513 ±0.14 |
| Exponential | DeepQuantReg | 20.046 ± 5.844 | 4.675 ±0.09 | 2.294 ±0.252 | 0.544 ±0.004 | 1.276 ±0.158 |
| Exponential | LogNorm MLE | 17.825 ± 2.831 | 4.342 ±0.058 | 1.056 ±0.071 | 0.552 ±0.003 | 0.401 ±0.045 |
| Weibull | CQRNN | **0.255 ± 0.025** | 1.879 ±0.018 | **0.33 ±0.037** | **0.772 ±0.002** | 0.211 ±0.023 |
| Weibull | Sequential grid | 0.261 ± 0.018 | **1.877 ±0.017** | 0.352 ±0.037 | **0.772 ±0.002** | **0.209 ±0.023** |
| Weibull | Excl. censor | 2.468 ± 0.118 | 2.18 ±0.021 | 1.942 ±0.152 | 0.769 ±0.002 | 0.834 ±0.069 |
| Weibull | DeepQuantReg | 1.116 ± 0.16 | 2.005 ±0.037 | 1.057 ±0.159 | 0.769 ±0.002 | 0.556 ±0.126 |
| Weibull | LogNorm MLE | 1.586 ± 0.081 | 2.048 ±0.02 | 0.603 ±0.036 | 0.771 ±0.002 | 0.478 ±0.072 |
| LogNorm | CQRNN | 0.411 ± 0.057 | 1.716 ±0.04 | 0.253 ±0.025 | 0.59 ±0.004 | 0.161 ±0.015 |
| LogNorm | Sequential grid | 0.328 ± 0.036 | 1.716 ±0.041 | 0.28 ±0.033 | **0.591 ±0.003** | 0.193 ±0.024 |
| LogNorm | Excl. censor | 1.757 ± 0.114 | 1.818 ±0.051 | 1.46 ±0.129 | 0.588 ±0.004 | 1.119 ±0.062 |
| LogNorm | DeepQuantReg | 1.279 ± 0.168 | 1.843 ±0.04 | 1.924 ±0.131 | 0.58 ±0.004 | 1.279 ±0.106 |
| LogNorm | LogNorm MLE | **0.247 ± 0.032** | **1.713 ±0.041** | **0.115 ±0.014** | 0.589 ±0.004 | **0.151 ±0.018** |
| Norm uniform | CQRNN | 0.388 ± 0.054 | 1.442 ±0.022 | 2.219 ±0.212 | 0.789 ±0.003 | **0.094 ±0.008** |
| Norm uniform | Sequential grid | **0.188 ± 0.016** | **1.409 ±0.017** | **1.347 ±0.136** | 0.788 ±0.003 | 0.462 ±0.02 |
| Norm uniform | Excl. censor | 1.088 ± 0.112 | 1.501 ±0.018 | 1.557 ±0.19 | **0.79 ±0.003** | 0.721 ±0.066 |
| Norm uniform | DeepQuantReg | 2.591 ± 0.56 | 1.809 ±0.084 | 4.397 ±0.687 | 0.598 ±0.064 | 0.418 ±0.074 |
| Norm uniform | LogNorm MLE | 939.992 ± 144.76 | 7.727 ±0.502 | 32.486 ±1.405 | 0.77 ±0.006 | 4.259 ±0.144 |
| Norm heavy | CQRNN | **0.579 ± 0.089** | 0.999 ±0.041 | **6.491 ±1.094** | 0.922 ±0.002 | 0.301 ±0.067 |
| Norm heavy | Sequential grid | 0.597 ± 0.081 | **0.996 ±0.037** | 6.595 ±1.406 | **0.923 ±0.002** | 0.198 ±0.014 |
| Norm heavy | Excl. censor | 1.285 ± 0.205 | 1.223 ±0.069 | 8.206 ±1.895 | **0.923 ±0.002** | 0.261 ±0.021 |
| Norm heavy | DeepQuantReg | 1.099 ± 0.175 | 1.166 ±0.06 | 6.681 ±1.365 | **0.923 ±0.002** | **0.191 ±0.016** |
| Norm heavy | LogNorm MLE | 5568.886 ± 1737.861 | 17.077 ±2.442 | 21.283 ±1.057 | 0.852 ±0.008 | 1.113 ±0.102 |
| Norm med. | CQRNN | **0.11 ± 0.007** | **0.789 ±0.006** | 0.633 ±0.138 | **0.896 ±0.001** | 0.157 ±0.054 |
| Norm med. | Sequential grid | 0.16 ± 0.009 | 0.799 ±0.005 | 0.474 ±0.046 | 0.895 ±0.001 | 0.159 ±0.011 |
| Norm med. | Excl. censor | 0.117 ± 0.008 | 0.792 ±0.005 | **0.247 ±0.02** | **0.896 ±0.001** | **0.136 ±0.015** |
| Norm med. | DeepQuantReg | 0.255 ± 0.016 | 0.847 ±0.008 | 0.944 ±0.098 | 0.892 ±0.001 | 0.232 ±0.029 |
| Norm med. | LogNorm MLE | 276.276 ± 41.622 | 3.974 ±0.228 | 17.969 ±0.548 | 0.865 ±0.004 | 4.612 ±0.134 |
| Norm light | CQRNN | **0.079 ± 0.005** | **0.778 ±0.005** | 0.173 ±0.021 | **0.882 ±0.001** | **0.084 ±0.008** |
| Norm light | Sequential grid | 0.117 ± 0.005 | 0.784 ±0.004 | 0.352 ±0.027 | **0.882 ±0.001** | 0.19 ±0.018 |
| Norm light | Excl. censor | 0.083 ± 0.005 | 0.779 ±0.005 | **0.159 ±0.017** | **0.882 ±0.001** | 0.112 ±0.013 |
| Norm light | DeepQuantReg | 0.277 ± 0.013 | 0.854 ±0.008 | 1.205 ±0.081 | 0.878 ±0.001 | 0.588 ±0.044 |
| Norm light | LogNorm MLE | 58.503 ± 6.922 | 2.475 ±0.093 | 13.713 ±0.465 | 0.861 ±0.003 | 7.74 ±0.286 |
| Norm same | CQRNN | **0.094 ± 0.005** | **0.785 ±0.003** | 0.19 ±0.022 | 0.893 ±0.001 | **0.052 ±0.007** |
| Norm same | Sequential grid | 0.779 ± 0.16 | 0.847 ±0.009 | 0.45 ±0.054 | 0.891 ±0.001 | 0.102 ±0.012 |
| Norm same | Excl. censor | 0.435 ± 0.01 | 0.927 ±0.007 | 4.096 ±0.251 | **0.894 ±0.001** | 1.398 ±0.07 |
| Norm same | DeepQuantReg | 0.357 ± 0.013 | 0.893 ±0.009 | 3.334 ±0.325 | 0.891 ±0.001 | 0.983 ±0.091 |
| Norm same | LogNorm MLE | 0.114 ± 0.008 | 0.787 ±0.004 | **0.187 ±0.024** | **0.894 ±0.001** | 0.059 ±0.006 |
| LogNorm heavy | CQRNN | 2.424 ± 0.055 | 1.123 ±0.021 | 22.493 ±0.36 | **0.782 ±0.005** | **0.036 ±0.004** |
| LogNorm heavy | Sequential grid | 2.42 ± 0.055 | 1.121 ±0.021 | 21.938 ±0.299 | 0.781 ±0.005 | 0.044 ±0.002 |
| LogNorm heavy | Excl. censor | 2.654 ± 0.061 | 1.247 ±0.021 | 35.43 ±0.629 | 0.772 ±0.005 | 4.806 ±0.226 |
| LogNorm heavy | DeepQuantReg | 2.639 ± 0.06 | 1.236 ±0.022 | 34.132 ±0.719 | 0.771 ±0.005 | 3.884 ±0.201 |
| LogNorm heavy | LogNorm MLE | **1.17 ± 0.052** | **0.868 ±0.018** | **0.135 ±0.014** | 0.766 ±0.005 | 0.074 ±0.008 |
| LogNorm med. | CQRNN | 1.713 ± 0.049 | 0.923 ±0.02 | 5.054 ±0.174 | **0.754 ±0.004** | **0.064 ±0.005** |
| LogNorm med. | Sequential grid | 1.699 ± 0.047 | 0.921 ±0.02 | 4.968 ±0.181 | **0.754 ±0.004** | 0.098 ±0.014 |
| LogNorm med. | Excl. censor | 2.168 ± 0.053 | 1.067 ±0.021 | 12.124 ±0.34 | 0.749 ±0.004 | 2.586 ±0.071 |
| LogNorm med. | DeepQuantReg | 2.087 ± 0.056 | 1.033 ±0.02 | 10.081 ±0.255 | 0.748 ±0.003 | 1.373 ±0.055 |
| LogNorm med. | LogNorm MLE | **0.907 ± 0.05** | **0.824 ±0.018** | **0.103 ±0.016** | 0.75 ±0.003 | 0.07 ±0.009 |
| LogNorm light | CQRNN | 0.506 ± 0.028 | **0.764 ±0.019** | 0.331 ±0.026 | **0.729 ±0.003** | 0.135 ±0.01 |
| LogNorm light | Sequential grid | 0.532 ± 0.029 | 0.767 ±0.018 | 0.517 ±0.04 | **0.729 ±0.003** | 0.21 ±0.014 |
| LogNorm light | Excl. censor | 1.185 ± 0.037 | 0.852 ±0.02 | 1.518 ±0.124 | **0.729 ±0.003** | 0.655 ±0.047 |
| LogNorm light | DeepQuantReg | 0.831 ± 0.036 | 0.804 ±0.019 | 0.912 ±0.061 | 0.726 ±0.003 | 0.438 ±0.028 |
| LogNorm light | LogNorm MLE | **0.432 ± 0.034** | 0.767 ±0.018 | **0.095 ±0.015** | **0.729 ±0.002** | **0.079 ±0.004** |
| LogNorm same | CQRNN | 0.137 ± 0.021 | 0.735 ±0.015 | 0.236 ±0.021 | 0.751 ±0.002 | 0.055 ±0.007 |
| LogNorm same | Sequential grid | 0.422 ± 0.054 | 0.753 ±0.016 | 0.463 ±0.046 | 0.752 ±0.003 | 0.101 ±0.018 |
| LogNorm same | Excl. censor | 1.068 ± 0.043 | 0.861 ±0.019 | 4.112 ±0.264 | 0.752 ±0.002 | 1.306 ±0.046 |
| LogNorm same | DeepQuantReg | 0.394 ± 0.057 | 0.763 ±0.016 | 1.301 ±0.265 | 0.748 ±0.002 | 0.335 ±0.072 |
| LogNorm same | LogNorm MLE | **0.067 ± 0.013** | **0.73 ±0.015** | **0.114 ±0.013** | **0.754 ±0.002** | **0.052 ±0.005** |

| Dataset | Method | MSE to true quantile (lower better) | Uncensored quantile loss (lower better) | Uncensored D-Calibration (lower better) | Concordance-Index (higher better) | Censored D-Calibration (lower better) |
|---|---|---|---|---|---|---|
| | | | **Type 2 – real data, synthetic censoring** | | | |
| Housing | CQRNN | - | **0.34 ±0.002** | **0.793 ±0.03** | 0.897 ±0.0 | **0.02 ±0.004** |
| Housing | Excl. censor | - | 0.443 ±0.005 | 2.176 ±0.057 | 0.895 ±0.001 | 0.311 ±0.011 |
| Housing | DeepQuantReg | - | 0.399 ±0.004 | 2.474 ±0.066 | **0.902 ±0.001** | 0.196 ±0.031 |
| Housing | LogNorm MLE | - | 0.6 ±0.002 | 2.794 ±0.022 | 0.881 ±0.001 | 1.035 ±0.015 |
| Protein | CQRNN | - | **0.435 ±0.001** | **0.275 ±0.008** | **0.847 ±0.001** | **0.027 ±0.001** |
| Protein | Excl. censor | - | 0.631 ±0.002 | 3.45 ±0.053 | 0.838 ±0.001 | 1.075 ±0.02 |
| Protein | DeepQuantReg | - | 0.568 ±0.002 | 2.809 ±0.059 | 0.831 ±0.001 | 0.495 ±0.011 |
| Protein | LogNorm MLE | - | 0.579 ±0.002 | 0.694 ±0.018 | 0.817 ±0.002 | 0.298 ±0.007 |
| Wine | CQRNN | - | **0.6 ±0.005** | **0.908 ±0.069** | **0.815 ±0.002** | **0.046 ±0.005** |
| Wine | Excl. censor | - | 0.791 ±0.005 | 6.606 ±0.209 | 0.799 ±0.003 | 0.722 ±0.038 |
| Wine | DeepQuantReg | - | 0.717 ±0.005 | 3.211 ±0.159 | 0.792 ±0.003 | 0.212 ±0.021 |
| Wine | LogNorm MLE | - | 1.454 ±0.022 | 5.736 ±0.163 | 0.747 ±0.004 | 0.784 ±0.022 |
| PHM | CQRNN | - | **0.408 ±0.001** | **0.243 ±0.012** | **0.902 ±0.001** | **0.008 ±0.001** |
| PHM | Excl. censor | - | 0.519 ±0.002 | 3.852 ±0.037 | 0.901 ±0.001 | 0.481 ±0.006 |
| PHM | DeepQuantReg | - | 0.479 ±0.002 | 1.589 ±0.057 | 0.897 ±0.001 | 0.154 ±0.011 |
| PHM | LogNorm MLE | - | 0.599 ±0.002 | 2.26 ±0.018 | 0.9 ±0.001 | 0.538 ±0.007 |
| SurvMNIST | CQRNN | - | **0.076 ±0.0** | **0.308 ±0.023** | 0.899 ±0.001 | **0.224 ±0.005** |
| SurvMNIST | Excl. censor | - | 0.133 ±0.002 | 2.115 ±0.086 | 0.896 ±0.001 | 0.512 ±0.014 |
| SurvMNIST | DeepQuantReg | - | 0.1 ±0.001 | 1.021 ±0.051 | **0.9 ±0.001** | 0.264 ±0.013 |
| SurvMNIST | LogNorm MLE | - | 0.209 ±0.001 | 4.348 ±0.049 | 0.894 ±0.001 | 0.806 ±0.015 |
| | | | **Type 3 – real data, real censoring** | | | |
| METABRIC | CQRNN | - | - | - | **0.644 ±0.006** | **0.189 ±0.057** |
| METABRIC | Excl. censor | - | - | - | 0.615 ±0.005 | 7.54 ±0.478 |
| METABRIC | DeepQuantReg | - | - | - | 0.601 ±0.006 | 2.393 ±0.211 |
| METABRIC | LogNorm MLE | - | - | - | 0.636 ±0.006 | 0.72 ±0.106 |
| WHAS | CQRNN | - | - | - | **0.85 ±0.005** | 1.089 ±0.431 |
| WHAS | Excl. censor | - | - | - | 0.785 ±0.006 | 9.391 ±0.709 |
| WHAS | DeepQuantReg | - | - | - | 0.774 ±0.008 | 6.71 ±0.448 |
| WHAS | LogNorm MLE | - | - | - | 0.81 ±0.005 | **0.817 ±0.139** |
| SUPPORT | CQRNN | - | - | - | 0.615 ±0.002 | **0.179 ±0.022** |
| SUPPORT | Excl. censor | - | - | - | 0.552 ±0.002 | 9.606 ±0.231 |
| SUPPORT | DeepQuantReg | - | - | - | 0.564 ±0.002 | 6.747 ±0.176 |
| SUPPORT | LogNorm MLE | - | - | - | **0.617 ±0.002** | 2.384 ±0.143 |
| GBSG | CQRNN | - | - | - | **0.683 ±0.005** | **0.339 ±0.033** |
| GBSG | Excl. censor | - | - | - | 0.673 ±0.005 | 11.732 ±0.512 |
| GBSG | DeepQuantReg | - | - | - | 0.673 ±0.005 | 8.998 ±0.479 |
| GBSG | LogNorm MLE | - | - | - | 0.677 ±0.004 | 0.793 ±0.08 |
| TMBImmuno | CQRNN | - | - | - | 0.579 ±0.008 | **0.201 ±0.04** |
| TMBImmuno | Excl. censor | - | - | - | 0.52 ±0.008 | 9.479 ±0.386 |
| TMBImmuno | DeepQuantReg | - | - | - | 0.539 ±0.011 | 4.827 ±0.306 |
| TMBImmuno | LogNorm MLE | - | - | - | **0.581 ±0.007** | 0.634 ±0.067 |
| BreastMSK | CQRNN | - | - | - | 0.619 ±0.01 | **0.08 ±0.012** |
| BreastMSK | Excl. censor | - | - | - | **0.646 ±0.008** | 4.546 ±0.235 |
| BreastMSK | DeepQuantReg | - | - | - | 0.638 ±0.008 | 3.0 ±0.17 |
| BreastMSK | LogNorm MLE | - | - | - | 0.631 ±0.01 | 0.521 ±0.058 |
| LGGGBM | CQRNN | - | - | - | 0.792 ±0.008 | 0.372 ±0.074 |
| LGGGBM | Excl. censor | - | - | - | 0.782 ±0.01 | 2.166 ±0.303 |
| LGGGBM | DeepQuantReg | - | - | - | 0.781 ±0.011 | 1.275 ±0.184 |
| LGGGBM | LogNorm MLE | - | - | - | **0.793 ±0.008** | **0.367 ±0.083** |

Table 5: Results comparing CQRNN and sequential grid algorithm on the type 3 datasets, real target data with real censoring. Experiments were repeated over 50 random seeds.

Raw Results, mean ± one standard error

| Dataset | Method | TQMSE | UQL | UnDCal | Concordance-Index (higher better) | Censored D-Calibration (lower better) |
|---|---|---|---|---|---|---|
| METABRIC | CQRNN | - | - | - | 0.643 ±0.003 | 0.218 ±0.026 |
| METABRIC | Sequential grid | - | - | - | 0.648 ±0.002 | 0.399 ±0.04 |
| WHAS | CQRNN | - | - | - | 0.86 ±0.002 | 0.721 ±0.091 |
| WHAS | Sequential grid | - | - | - | 0.852 ±0.002 | 5.038 ±0.74 |
| SUPPORT | CQRNN | - | - | - | 0.614 ±0.001 | 0.159 ±0.01 |
| SUPPORT | Sequential grid | - | - | - | 0.613 ±0.001 | 0.723 ±0.024 |
| GBSG | CQRNN | - | - | - | 0.678 ±0.002 | 0.361 ±0.024 |
| GBSG | Sequential grid | - | - | - | 0.678 ±0.002 | 0.789 ±0.042 |
| TMBImmuno | CQRNN | - | - | - | 0.571 ±0.003 | 0.207 ±0.021 |
| TMBImmuno | Sequential grid | - | - | - | 0.572 ±0.003 | 0.375 ±0.028 |
| BreastMSK | CQRNN | - | - | - | 0.618 ±0.005 | 0.085 ±0.01 |
| BreastMSK | Sequential grid | - | - | - | 0.597 ±0.006 | 0.227 ±0.016 |
| LGGGBM | CQRNN | - | - | - | 0.784 ±0.004 | 0.397 ±0.039 |
| LGGGBM | Sequential grid | - | - | - | 0.781 ±0.004 | 0.491 ±0.041 |

Difference in means, alongside 95% confidence interval

| Dataset | Number quantiles | Training time speed up | Test time speed up | Parameter saving | C-Index difference Seq. grid - CQRNN (>0 favours CQRNN) | CQRNN is sig. better? | CensDCal difference Seq. grid - CQRNN (<0 favours CQRNN) | CQRNN is sig. better? |
|---|---|---|---|---|---|---|---|---|
| METABRIC | 5 | 5.3× | 2.5× | 4.6× | -0.005 ± 0.001 | ✗ | -0.181 ± 0.032 | ✓ |
| WHAS | 5 | 5.1× | 3.9× | 4.6× | 0.008 ± 0.001 | ✓ | -4.317 ± 0.745 | ✓ |
| SUPPORT | 5 | 5.1× | 2.7× | 4.6× | 0.001 ± 0.000 | ✓ | -0.564 ± 0.022 | ✓ |
| GBSG | 5 | 5.3× | 5.1× | 4.6× | -0.001 ± 0.000 | ✗ | -0.428 ± 0.026 | ✓ |
| TMBImmuno | 5 | 5.1× | 4.3× | 4.6× | -0.000 ± 0.001 | – | -0.168 ± 0.025 | ✓ |
| BreastMSK | 5 | 5.1× | 5.0× | 4.6× | 0.021 ± 0.004 | ✓ | -0.141 ± 0.018 | ✓ |
| LGGGBM | 5 | 5.4× | 5.1× | 4.6× | 0.003 ± 0.001 | ✓ | -0.094 ± 0.038 | ✓ |
| CQRNN better: | | | | | | 4/7 | | 7/7 |
| Seq grid better: | | | | | | 2/7 | | 0/7 |
| No sig. difference: | | | | | | 1/7 | | 0/7 |

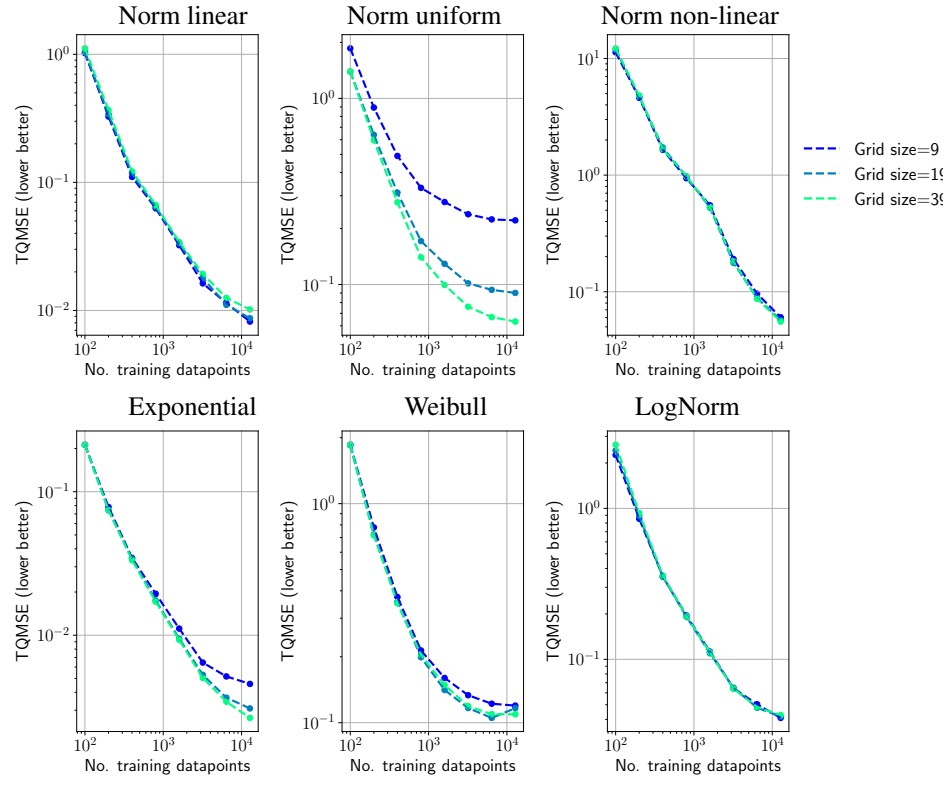

Figure 4: Ablations over grid size and number of datapoints for various synthetic datasets using our CQRNN method.

## C.1 Hyperparameter ablations

The requirements for CQRNN in Algorithm 2 include two hyperparameters unique to CQRNN (the rest determine the NN and its optimisation) – the grid of quantiles, $\mathrm{grid}_\tau$, and the large pseudo y value, $y^*$. This section investigates the role of these. It also discusses two modifications which were tested in initial experiments but which were not deemed essential for good performance, and excluded from the proposed method – mitigating crossing quantiles and interpolating between quantiles.

**Grid size.** For evenly spaced grids, as considered in this work, the number of quantiles estimated, $M$, fully determines the grid. One might expect that a larger grid size, with finer increments between quantiles, would lead to a better fit, since censored weights can be estimated more accurately. Indeed, the convergence rate of Portnoy's estimator was found to depend on grid size, $\mathcal{O}(1/(MN))$ [Neocleous et al., 2006].

We hypothesised that a larger grid might only deliver benefit when the dataset was sufficiently large for these fine-grained quantiles to be distinguished. Hence, Figure 4 shows an ablation investigating the interaction between grid size and number of datapoints for each of our type 1 synthetic 1D functions. Experiments were repeated with 100 random seeds, which was required to reduce the error bars sufficiently for comparison. Grid size was varied, $M \in \{9, 19, 39\}$, and number of datapoints, $N \in \{100, 200, 400, 800, 1600, 3200, 6400, 12800\}$.

For all datasets and grid sizes, TQMSE decreases with dataset size. In general a larger grid size does produce lower TQMSE, and in three datasets the benefit is significant (Norm uniform, Exponential, Weibull), with the advantage indeed more pronounced with a larger dataset size. In two datasets (Norm non-linear, LogNorm), this benefit is slight, and a larger grid is even seen to be slightly harmful on very small datasets. One dataset (Norm linear) does not follow this trend, where the widest grid proves slightly harmful across all dataset sizes.

**Pseudo y value.** Portnoy [2003] proposed that $y^*$ could be set to any large value approximating $\infty$, with the R package 'quantreg' setting $y^* = 1e6$. Since the sequential grid algorithm learns quantiles sequentially, it can simply halt if it attempts to estimate a quantile for which only censored datapoints remain, and hence is undefined.

The CQRNN algorithm changes this situation in two ways. Firstly, since the algorithm is no longer sequential and learns all quantiles simultaneously, it does not have the option of halting. Secondly, since a non-linear function is learnt, it is possible that higher quantiles might be undefined in one region of the input space, whilst being learnable in the rest of the space. Regressing towards an arbitrarily large $y^*$ value for a portion of the input space could adversely impact the quantile estimate elsewhere. When all quantiles in $\mathrm{grid}_\tau$ are fully defined, the effect is no different to using $\infty$.

To accommodate these differences, we recommend setting $y^*$ to a more modest value. We define it in terms of the maximum y value in the training set, $y^* = c_{y^*} \max_i y_i$, for a hyperparemeter, $c_{y^*} > 1$. In real-world problems, a practitioner might use an estimate for this based on their knowledge about the maximum feasible value for the target. In lieu of that, we set $c_{y^*} = 1.2$ for all our experiments (except the below ablation!), which provided reasonable results across datasets.

Table 6 presents an ablation on four of our (multidimensional) type 1 synthetic datasets, using $c_{y^*} \in \{1.0, 1.2, 1.5, 2.0, 10, 9, 100.0\}$. For dataset Norm light, $y^*$ has no impact since the target distribution is fully defined under censoring. However, other datasets have input regions where the higher quantiles are *not* defined due to censoring, and hence higher quantiles are impacted by the magnitude of $y^*$. The optimal value varies by dataset (e.g. 1.2 is best for Norm heavy, while 10.0 is best for LogNorm heavy).

Table 6: Ablation over psuedo y value, $y^*$. Mean over ten runs, all hyperparameters fixed.

| Dataset | $1.0\max_i y_i$ | $1.2\max_i y_i$ | $1.5\max_i y_i$ | $2.0\max_i y_i$ | $10.0\max_i y_i$ | $100.0\max_i y_i$ |
|---|---|---|---|---|---|---|
| | | | **TQMSE (lower better)** | | | |
| Norm heavy | 1.452 | 0.579 | 1.237 | 6.196 | 591.257 | 57421.824 |
| Norm light | 0.081 | 0.079 | 0.081 | 0.081 | 0.081 | 0.081 |
| LogNorm heavy | 2.502 | 2.424 | 2.321 | 2.173 | 1.026 | 19.827 |
| LogNorm light | 0.614 | 0.506 | 0.385 | 0.252 | 0.147 | 0.147 |

**Crossing quantiles.** One issue often discussed in quantile regression is 'crossing quantiles'. Higher quantiles should *always* produce higher predictions, i.e. $\hat{y}_{i,\tau_1} > \hat{y}_{i,\tau_2} \forall \tau_1 > \tau_2$. The crossing quantile

issue arises when this condition does not hold. We anticipated that the flexibility of NNs might exacerbate this issue, and we tested two methods to remedy this. 1) Adding a crossing penalty to the loss [Bondell et al., 2010], $\mathcal{L}_{\text{cross}} = \sum_{i=1}^{N} \sum_{j=1}^{N_\tau - 1} \max[0, c - (\hat{y}_{i,\text{grid}_\tau[j+1]} - \hat{y}_{i,\text{grid}_\tau[j]})]$, where $c$ is the smallest acceptable distance between neighbouring quantiles. 2) Modifying the NN architecture to enforce monotonicity between quantiles, by constraining each consecutive quantile prediction to add on to the previous one, after passing through a SoftPlus. In our experiments, neither method significantly impacted performance. Favouring simplicity, we propose the CQRNN algorithm without these. It's possible that other methods might have more effect, e.g. [Zhou et al., 2020, Brando et al., 2022]. We leave further exploration to future work.

**Interpolating quantiles.** The CQRNN algorithm sets the estimated censored quantiles, **q**, by choosing the prediction closest to the censored datapoint, $\hat{q}_j \leftarrow \arg\min_\tau |\hat{y}_{j,\tau} - y_j|$. In early experiments, we considered an alternative approach that took a linear interpolation between the two nearest quantiles. In initial experiments this wasn't found to significantly improve performance, so we propose the CQRNN algorithm using the simpler $\arg\min$ approach.

**Partial vs. full optimisation.** Figure 5 explores the effect of optimising the NNs partially, as is proposed in CQRNN in Algorithm 2, compared to fully, as might be done more typically in EM procedures. The figure shows that convergence is fastest when using partial maximisation, when the most up-to-date estimates of $\hat{q}_i$ are used. This ends up being more efficient than freezing $\hat{q}_i$ and only updating after a longer period of optimisation.

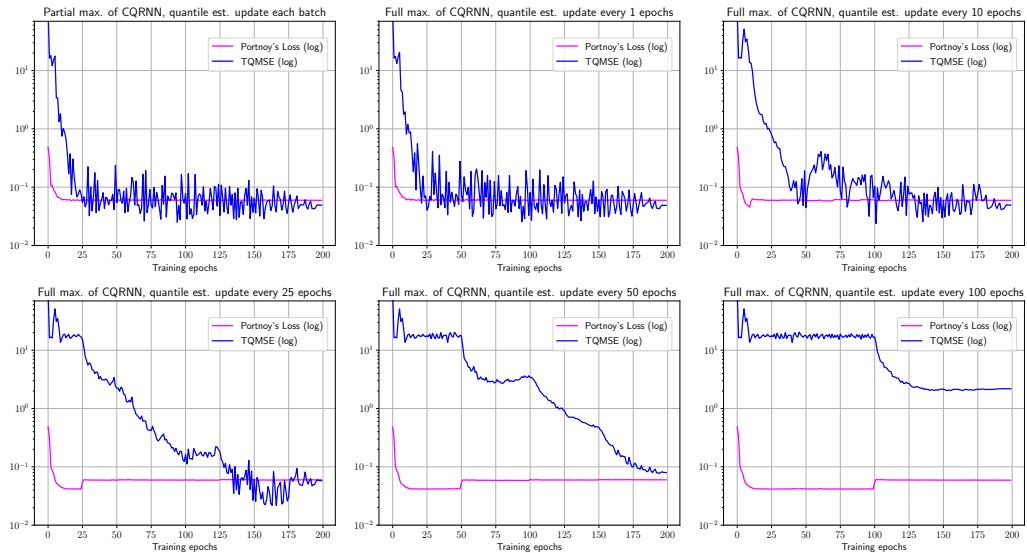

Figure 5: This figure explores the effect of optimising the NNs partially compared to fully. It shows training loss and TQMSE over training epochs. The Normal Linear dataset was used.