# OpenReview forum: "Censored Quantile Regression Neural Networks for Distribution-Free Survival Analysis"
_NeurIPS.cc/2022/Conference — NeurIPS 2022 Accept_

### Official Review · Reviewer_qGAg · 2022-07-03

**Rating:** 6
**Confidence:** 4
**Soundness:** 2 fair
**Presentation:** 3 good
**Contribution:** 2 fair

**Summary:**

The authors use neural networks to replace the linear models in censored quantile regression. And the authors further improve the sequential grid algorithm with bootstrap weights. The authors explain the soundness of their method from an analogy of EM. The authors compare their method with three baselines on synthetic and real dataset.

**Questions:**

1. Can we get a more concrete guarantee on the convergence of the propose algorithm? Some guarantee when the model is linear should also be good.
2. Quantile regression seems to give us quantiles for certain $\tau$'s . But for a valid survival model, we need the whole distribution. How do we get the whole distribution? Or why do we not need to whole distribution in survival modeling? In Portnoy (2003), $\tau$ seems to be part of the function so we do not infinite amount of $\tau$'s. But in the authors' setting, for a different $\tau$, we need a different model. This requires the infinite amount of $\tau$'s to get the whole distribution.
3. Why do authors choose lognormal model as the baseline for survival modeling?

**Limitations:**

One extra limitation seems to be that CQRNN could not get the full survival distribution.

**Strengths And Weaknesses:**

Strengths:
1. The method is novel. Introducing nn into quantile regression and the new bootstrap algorithm are both contributions to the society.
2. The authors make some effort to explain why and how the bootstrap algorithm CQRNN works.
3. The writing is clear. It's easy for me to follow the paper.

Weakness:
1. I appreciate that the authors explain how CQRNN work from an EM analogy and self-correcting property. But they still sound vague to me. Can we have a simpler example to understand it in a more rigorous way? For example, if the model is linear, how the convergence property is like using the CQRNN bootstrap algorithm?
2. The empirical comparison does not seem to be thorough if the authors want to prove that their model is a good "survival model".
- On synthetic datasets, the authors only compare the performance only on several $\tau$'s (0.1, 0.5, 0.9). But for a survival model, why do we only care about certain $\tau$'s? How do we get values at other $\tau$'s that the quantile regression has not modeled? For a survival model, quantile at certain $\tau$'s is not the only thing we care about. We may need to look at the whole survival distribution. And then we compare other scores, for example, Brier score.
- The authors only consider lognormal model as their baseline and the reason is that they believe "this distribution often suits properties of real-world time-to-event survival data." Why do you believe so? As far as I know, lognormal models have strict distributional assumptions and may not have good performance. It is also hard to train due to the exp transformation from normal. Other models like DeepHit, SODEN may have better performance on concordance or calibration.

---

> ### Author Response · Authors · 2022-08-02
> **Author Response to Reviewer qGAg**
>
> Thank you for taking the time to review our paper. We are pleased you see novelty in our work, and found it easy to follow. Below, we have clarified the goals and utility of quantile regression, and respond to your specific concerns. We apologise if the paper was not clear on these points -- we have begun updating sections to rectify this. We hope that our comments help highlight the value of our contributions, and may warrant a re-evaluation of your score.
>
> __Quantile regression vs. full distribution modelling.__
> Quantile regression allows prediction of a distribution at a finite set of quantiles. A major benefit is that one avoids assuming any specific form of distribution between these quantiles. A drawback is that one has access to this distribution only at this fixed set of quantiles, and not to the full distribution. We agree with you that for certain problems, predictions at this set of fixed quantiles may not be sufficient, and a full predictive distribution should be sought -- here we may not recommend CQRNN.
>
> Nevertheless, having predictions at multiple quantiles can be more useful than having a point estimate prediction, since one has a measure of the uncertainty. This is often summarised as a prediction interval, say between $\tau = 0.1$ and $\tau = 0.9$.
> For example, knowing a patient has 10 years $\pm 1$ year to live, compared to knowing they have 10 years $\pm 8$ years, might greatly impact decisions around medical care and financial planning.
>
> It is with prediction intervals and point estimates in mind that we chose to evaluate many metrics at $\tau \in [0.1, 0.5, 0.9]$. These $\tau$ values were available across all grids we tested, allowing comparison of model quality across different values of $M$.
> Our metrics on calibration are assessed across all quantiles.
>
> It is incorrect to say that Portnoy’s method allows access to an infinite set of $\tau$'s. Their method is equivalent to our sequential grid algorithm, which uses a predefined fixed set of $\tau$'s.
> However, if one is determined to move from the set of predefined quantile predictions to predicting at intermediate quantiles, an obvious approach is to linearly interpolate between two adjacent predicted quantiles. This technique can be used with both CQRNN and Portnoy's linear models. (We touch on this in B.1.) But doing this implicitly makes a distributional assumption, which was what quantile regression tried to avoid!
>
> __Using Log Normal MLE as a baseline.__
> Survival data typically have skewed distributions with long tails which suit certain parametric distributions. From a popular textbook: "Examples of distributions that are commonly used for survival time are: the Weibull, the exponential, the log-logistic, the log-normal, and..." [4] (page 292). Hence, we said Log Normal "suits properties of real-world time-to-event survival data".
>
> LogNormal MLE has been used by recent works on NNs and survival analysis as a baseline [1, 2, 3], which we believe made it important to for us to include. We felt it was interesting to compare the distribution-free approach taken by CQRNN, with a model that commits to an explicit form of distribution. We did not run into trouble with its optimisation.
>
> We agree with you that other methods may also make relevant baselines. We focused most of our efforts on comparing to quantile-based methods, which are aligned with CQRNN's objective. If you feel strongly about including a particular baseline, let us know and we can try to implement this. Thanks also for making us aware of SODEN, which we were unfamiliar with, and have cited.
>
> __Convergence guarantees for CQRNN.__
> We did attempt to derive convergence guarantees for linear models, but this turns out to be difficult!
> A challenge in analysing CQRNN is that there is a complex feedback loop between the quantile estimates, which affect the loss function, which affects the optimisation, which affects the new quantile estimates...
> Hence, even when model is linear, the algorithm's behaviour is not.
> So while we could study how individual quantile estimates, $\hat{y}_{j, \tau}$, moved during optimisation given weights (Theorem 2), we were unable to analyse how these movements affect future iterations of the algorithm. The EM analogy does provide insight into how this feedback works.
> Since the objective of our paper is showing applicability of the algorithm to NNs and not linear models, we avoided committing too much effort to studying the linear case. The EM analogy and the self-correcting property presented both hold for general non-linear models.
>
> [1] X-CAL: Explicit calibration for survival analysis, Goldstein et al., NeurIPS 2020
> [2] Countdown regression: Sharp and calibrated survival predictions, Avati et al., UAI 2019
> [3] Modeling Progression Free Survival in Breast Cancer with Tensorized Recurrent Neural Networks and Accelerated Failure Time Models, Yang et al., 2017
> [4] Survival Analysis, A Self-Learning Text, Kleinbaum \& Klein, 2012

---

> > ### Comment · Reviewer_qGAg · 2022-08-06
> > **The authors' feedback did not solve all my concerns but convinced me; raised my score**
> >
> > I thank the authors for detailed replies. The weakness in my review are still there. But I think the novelty of the paper outweighs the weakness. So I raised my score. I thank the authors for the clarification but the problem is still there.
> >
> > 1. Using Log Normal MLE as a baseline. Adding more baselines will make your work more convincing. But I am okay with lognormal as a baseline as what you said it's a baseline that has distributional assumptions.
> >
> > 2. Convergence guarantees for CQRNN. I understand this is hard. But this makes me not giving a high score and makes the algorithm mysterious.
> >
> > 3. Quantile regression vs. full distribution modeling. The reply is convincing.

---

> > > ### Author Response · Authors · 2022-08-08
> > > **Reply**
> > >
> > > Thank you for your response and updated score. We really appreciate your open-mindedness in this process.

---

### Official Review · Reviewer_57zr · 2022-07-06

**Rating:** 6
**Confidence:** 4
**Soundness:** 3 good
**Presentation:** 3 good
**Contribution:** 2 fair

**Summary:**

The paper considers the task of quantile regression in a survival analysis context, i.e. one where some examples are right-censored. As in almost all recent works, the encoder backbone is a neural network. The authors start by adapting the sequential grid (SG) algorithm to work with neural networks, as opposed to strictly linear models. Noting the inneficiency of the adaptation, they propose CQRNN, a novel algorithm which is more efficient. CQRNN departs from SG in that the assignment weights are updated during training.
In addition to introducing their proposed algorithm and presenting experimental results jusitying it's adoption, the authors provide a qualitative discussion of their algorithm. In particular, they frame it as an Expectation-Maximization algorithm, and provide a proof sketch as to why their proposed solution should result in "sensible" weight allocations.

**Questions:**

- Could the authors re-do the comparison in table 1? I would like to confirm whether the purported advantage for their method is statistically significant.
- Would it make sense to adapt classical survival models (the authors cite a few in their work) to this context by simply changing the target to be the quantile center? I am curious what would be the relative performance of these approaches: quite a few are very strong in their original field.

Edit: The authors have responded to my questions. Taking this into account, I am raising my score.

**Limitations:**

I don't see any potential negative societal impact for this work. If anything, survival analysis is a field that has the potential to *positively* impact medicine.

**Strengths And Weaknesses:**

I. Strenghts
- The paper is well-written, and reasonably clear. The discussion is easy to follow, and the relevant prior works necessary to understand the novel approach are made explicit.
- The algorithm is clearly presented (good explanations + an explicit algorithm).
- Experiments are well detailed: hyperparameter optimization of the competing methods was done (to a certain extent).

II. Weaknesses.
- As the authors note, the performance gains over vanilla sequential grid are minor (I am referring to table 1). In particular, it is hard to tell whether they are statistically significant without error bars or similar information. As the difference between the two is a major point (otherwise the contribution is incremental), the question I ask in the next section has the goal to clarify this point.

---

> ### Author Response · Authors · 2022-08-02
> **Author Response to Reviewer 57zr**
>
> Thank you for your review. Allow us to respond to your queries below. We have updated the results table at your request with error bars and run a test of statistical significance. We have additionally extended this analysis to the real datasets. We hope this demonstration of CQRNN offering statistically significant improvements might justify an increase in your evaluation of the paper.
>
> __Performance gains of CQRNN vs. sequential grid (SG).__
> Thanks for your suggestion, and please accept our apologies for not including error bars originally. We have repeated the experiments in Table 1 (highlights five synthetic datasets) and Table 5 (which reports all 14 synthetic datasets) and reported 95\% confidence intervals in the updated paper version. CQRNN significantly outperforms SG on 8/14 datasets, while SG significantly outperforms CQRNN on 3/14 datasets, and there is no statistically significant difference (when the 95\% confidence intervals include 0.0) on 3/14 datasets. One instance of lower CQRNN performance is due to a large number of censored datapoints falling below the first quantile (see $^*$), and disappears with a larger grid size $M$.
>
> In addition to this comparison on the synthetic type 1 datasets, we felt it was important to evaluate whether CQRNN's performance boost also held on real data. We therefore have also run a head-to-head comparison of CQRNN vs. SG on type 3 datasets, included in our brand new Table 6. In terms of C-Index, CQRNN outperforms SG on 4/7 datasets while SG outperforms CQRNN on 2/7 datasets. In terms of calibration, CQRNN outperforms SG on all 7/7 datasets.
>
> We believe that this improvement in quantile accuracy, combined with an order of magnitude saving in terms of training time, test time, and parameter count, makes CQRNN preferable to SG for any researcher or practitioner applying quantile regression using NNs, to survival data. Please let us know whether you agree with this belief, and if not, whether you would like to see any more experiments as further evidence?
>
> __Contribution of applying the sequential grid algorithm to NNs.__
> Whilst applying Portnoy’s estimator to NNs might seem an obvious thing to try from the way the paper is presented, this has not previously been done, and we wanted to ensure our paper is credited for this contribution, as well as development of CQRNN. A lot of our research effort was spent combing the survival literature and experimenting with the various estimators to select one that could be adapted to NNs. Even the SG algorithm we used (presented in Algorithm 1) was modified from the original paper. Specifically, we make the initialisation scheme more efficient than the original version, which is supported by our analysis in section C.2.
>
> __Adapting classical survival models.__
> We are in agreement that there are many great ideas in the classical survival literature, that could help guide NN design for survival analysis, and might also be combined with ideas from quantile regression.
> Feel free to reach out after the review process if you had a specific idea in mind to explore!
>
>
>
> $^*$We investigated the dataset where SG outperforms CQRNN by a large margin (Norm Uniform), finding that heavy censoring below the first quantile is the cause of the performance difference. SG initialises it’s quantile estimates $\hat{\mathbf{q}} \gets 0.0$ (Algorithm 1), while the lowest available value for CQRNN $\hat{q}_i $ is 0.1 when $M=9$ (since, $\operatorname{grid}_\tau \in [0.1,0.2 ... 0.9]$). In the Norm Uniform dataset, most censored datapoints are closer to the 0.0 quantile than 0.1 (see Figure 3), so SG has a slight advantage. When $M$ is increased to 19, meaning the lowest available quantile value for CQRNN is 0.05, the performance of CQRNN and SG is not statistically different (included as an extra row in Table 5).

---

> > ### Comment · Reviewer_57zr · 2022-08-08
> > **Response to author reply**
> >
> > I thank the authors for taking the time to respond to my questions. Their answer is detailed, and adresses my main reservation (as outlined in my initial review). As a result of this, I am raising my score to a 6 (from a 5 initially).

---

> > > ### Author Response · Authors · 2022-08-08
> > > **Reply**
> > >
> > > Thank you for taking the time to go over our rebuttal. We are delighted to hear we were able to address your main reservation.

---

### Official Review · Reviewer_doRh · 2022-07-11

**Rating:** 7
**Confidence:** 3
**Soundness:** 4 excellent
**Presentation:** 3 good
**Contribution:** 3 good

**Summary:**

This paper proposes a neural network based approach for doing quantile regression on possibly censored data, i.e. when the target variable is sometimes not directly observed and instead a lower or upper bound is known, as in the case of survival data.
The authors first build on the linear approach on Portnoy 2003 and extend it straightforwardly to neural networks. However this approach requires the sequential optimization of a new NN for each quantile level to be predicted, which is computationally heavy.
Then a new method is proposed for simultaneous quantile regression , which is interpreted as a form of expectation maximization, allowing to estimate all the quantiles simultaneously.


**Questions:**

I think it would be interesting to discuss and relate this work to the recent literature on simultaneous quantile regression with NN that alleviates or avoids the crossing quantile problem. In particular, it would be interesting to discuss how these methods could be adapted to censored data using your construction.

A non-exhaustive list:
- *Tagasovska, N., & Lopez-Paz, D. (2019). Single-model uncertainties for deep learning. Advances in Neural Information Processing Systems, 32.*
- *Zhou, F., Wang, J., & Feng, X. (2020). Non-crossing quantile regression for distributional reinforcement learning. Advances in Neural Information Processing Systems, 33, 15909-15919.*
- *Brando, A., Center, B. S., Rodriguez-Serrano, J., & Vitria, J. (2022, May). Deep Non-Crossing Quantiles through the Partial Derivative. In International Conference on Artificial Intelligence and Statistics (pp. 7902-7914). PMLR.*

Minor comments:
Algorithm 1: since $\tau \leftarrow grid_\tau[i-1]$,  $\hat{\boldsymbol{q}}[K]$ and $\hat{\boldsymbol{q}}[\neg K_{cross}]$ are assigned the same value, is this correct?

Algorithm X, Equation X, Appendix X,… should start with a capital letter

To avoid confusion, I think it is better to say quantile level for $q_j$

Lines 187-191: $\bar{w}_j$ is the true weight, $\hat{w}_j$ is the estimate, what is $w_j$ then?

Line 324: which were the methods use to combat the crossing-quantile problem?



**Limitations:**

Yes

**Strengths And Weaknesses:**

Strengths:
- The paper is clearly written and the background on censored data and survival analysis is properly explained.
- The analysis gives interesting insights and useful guarantees on the algorithm
- The experiments are interesting and appropriate.

Weakness:
- Lacks a discussion on the recent literature on simultaneous quantile regression with NN

---

> ### Author Response · Authors · 2022-08-02
> **Author Response to Reviewer doRh**
>
> Thank you for your positive review, and particularly your attention to detail. We respond to your comments below.
>
> __Discussion of quantile regression with NNs.__
> We agree that our related work section was missing a section covering quantile regression and NNs, and have added this to the paper.
> We have included your suggested citations.
>
> As we understand, CQRNN is compatible with those crossing quantile remedies. We extended our discussion of this in appendix B.1, which describes the two methods we already trialled. We remain surprised that these crossing-quantile methods did not deliver empirical gains on our assessed metrics, and hope this might be explored in future work.
>
> __Algorithm 1, $\tau$ query.__
> Thanks for pointing this out, actually the line $\tau \gets \operatorname{grid}_\tau[i-1]$ should have been written $\tau \gets \operatorname{grid}_\tau[i]$ (otherwise this throws an error when $i=0$). We have updated this.
>
> __Capitalisation of references.__
> We note the NeurIPS style guide also recommends capitalising these words, and have updated the paper with this.
>
> __$w_j$ notation.__
> We introduced, $\bar{w}_j$ \& $\hat{w}_j$, in order to distinguish from $w_j$, which is used in a more general sense throughout the text to refer to censored weights without being specific if it refers to estimates or ground truth. Our notation also follows Portnoy’s own (their Eq. 14, 16). We have not edited this in the paper for now -- let us know if you have a preferred notation scheme.

---

> > ### Comment · Reviewer_doRh · 2022-08-09
> > **Notation**
> >
> > Any notation is fine if it is consistently used.
> > When you write $w_j = \frac{\tau - q_j}{1- q_j}$ , you say that $q_j$ is the quantile at which the data point was censored, so I understand that $q_j$ it is the true quantile, so $w_j$ should be the true weight.
> > In Algorithm 1, the estimate $\hat{\bf{q}}$ is used to compute $\bf{w}$, so here $\hat{\bf{w}}$ would make more sense since it is an estimate too.
> > Therefore, I think that the additional notation $\bar{w}_j$ for the true $w_j$ is not really needed.
> > When you treat the weights as latent variables instead of a fixed unknown parameters (Section 4), I think you can still use $\bf{w}$.

---

> > > ### Author Response · Authors · 2022-08-09
> > > **Reply**
> > >
> > > Thanks for pushing on this. After another look, we agree that Algorithm 1 \& 2 should use $\hat{\mathbf{w}}$ (it conflicts as is), and it would be better to be consistent in the text about whether we refer to estimates or true quantities (e.g. line 117 should use estimates). As per your suggestion, dropping the bar notation, $\bar{w}_j \to w_j$, seems the simplest way to resolve this. We will update this notation in the next version of the paper.

---

### Official Review · Reviewer_mH2a · 2022-07-14

**Rating:** 7
**Confidence:** 5
**Soundness:** 3 good
**Presentation:** 3 good
**Contribution:** 3 good

**Summary:**

The authors proposed a neural network approach for quantile regression on censored data, based on Portnoys's estimator. The authors also proposed a new algorithm to mitigate the computational challenge when combining neural networks and the existing sequential grid algorithm for Portnoys's estimator. In addition, the authors made the connection between the proposed CQRNN algorithm and the EM algorithm for further understanding and compared CQRNN with existing methods on both synthetic and real-world datasets.

**Questions:**

Please see the weaknesses above.

**Limitations:**

Yes, the authors discussed the limitations.

**Strengths And Weaknesses:**

Strengths:
* The proposed method is not just replacing the linear model with a neural network in the existing algorithm for Portnoy's estimator. They did identify new computational challenges when optimizing a neural network through the existing sequential grid algorithm, which was less problematic for linear models, and proposed a new CQRNN algorithm that achieves faster speeds and saves memory (while the improvement mainly depends on the size of the quantile grid).
* The connection to the EM algorithm under certain distribution assumptions and the self-correcting property are very interesting. I have a related question below.
* The authors provided solid numerical evaluations and comparisons with existing approaches both qualitatively and quantitatively, and in terms of both prediction accuracy and computation efficiency.


Weaknesses:
* It would be worthwhile to add more discussion in "related work" so as to explicitly highlight the improvement or limitation in comparison to other existing methods.
* For partial maximization, the authors mentioned the alternative version that is closer to the standard EM algorithm. While in each iteration the proposed algorithm saves computational costs by taking a single gradient step, the number of iterations required for convergence might also change or even increase. Do the authors have any comments on this or have the authors investigated the impact of taking a single gradient instead of multiple gradient descents on the overall computing time/convergence?
* From Table 2, the censoring distribution in LogNorm (and Norm uniform, etc.) in synthetic datasets seems to be independent of covariates. In this case, the assumption in DeepQuantReg also holds while CQRNN still performs better than DeepQuantReg. Can the authors explain this?

---

> ### Author Response · Authors · 2022-08-02
> **Author Response to Reviewer mH2a**
>
> Thank you for your review, we are pleased you were able to understand our work with full clarity and appreciated the value of our contributions. We respond to your questions below.
>
> __Extend related work to highlight pros and cons of CQRNN.__
> We have expanded the related work section to discuss when CQRNN might be preferred compared to parameterised distributions (e.g. our LogNorm MLE baseline), and the advantages and disadvantages of CQRNN compared to censored quantile linear models. Please let us know if you feel any further discussion might be helpful.
>
> __Investigation of the effect of partial vs. full maximisation.__
> This is a very interesting point. In fact we originally anticipated that the algorithm would have to fully optimise the NN before updating the $\hat{q}_i$’s, and repeating. Initial experiments then suggested it was possible to update the $\hat{q}_i$’s online, after partial maximisation. This seemed to give fast convergence, was stable, and it also resulted in a simpler algorithm, which encouraged us to pursue this direction.
>
> Inspired by your comment, we have added plots in appendix Figure 5 (also text in Section B.1 and 4.1), comparing partial optimisation as in Algorithm 2, with variants that do a fuller optimisation before updating quantile estimates, $\hat{q}_i$. These results were run on our synthetic Normal Linear dataset. We observe that the partial optimisation as presented in CQRNN converges fastest.
>
> __Performance of DeepQuantReg when censoring is independent of covariates.__
> Another very good point! We spent some time investigating. One potential issue we noticed is that the estimator from Huang et al. [1] (which DeepQuantReg builds on) is designed for recovering only the median, $\tau=0.5$, but DeepQuantReg directly applies it to other quantiles. We suspect there might be some undesirable bias introduced when applying this to quantiles away from the median.
> In contrast, Portnoy's estimator is consistent across quantiles. This could explain the difference in performance of CQRNN and DeepQuantReg even when censoring is independent of covariates.
> Curiously, we also noted that [1] states they do not require independence of covariates and censoring.
>
>
> More generally, Huang's and Portnoy's estimators are quite different approaches, with the former weighting observed datapoints only, while the latter splits censored data into two pseudo datapoints, and this in itself could lead to differing results.
>
> [1] Least Absolute Deviations Estimation for the Accelerated Failure Time Model, Huang et al. 2007

---

### Author Response · Authors · 2022-08-02
**Rebuttal, Summary of Paper Changes**

Thanks to all of you for your insightful reviews, which have guided us in improving the paper substantially. We have now uploaded a new version (main and appendix now joined in one pdf). Notable changes include:
* Improved comparison of CQRNN with the sequential grid. We've refreshed Table 3 and 5, repeating over multiple random seeds, have added error bars, and implemented tests of statistical significance. We've also added similar analysis on real datasets in Table 6.
* We have included new analysis in Appendix Figure 5, comparing the partial optimisation performed by CQRNN, with a full optimisation procedure that is closer to the typical EM algorithm.
* Related work has been significantly extended. Thanks to you all for pointing us to important prior work.

We have responded to each of your reviews individually. We are happy to engage in continued discussion during the discussion period.

---

### Meta-Review · Area_Chair_Ge1S · 2022-08-25

**Recommendation:** Accept
**Confidence:** Certain

**Metareview:**

This paper studies the quantile regression of censored data. Neural network models are used as the statistical model. Numerical results show the proposed algorithm is computationally efficient and attains high prediction accuracy compared to existing methods. Since reviewers agree that this paper is well written and interesting, I recommend accepting the paper.

**Award:**

No

---

### Decision · Program_Chairs · 2022-09-14

Accept